# Comparative Genomics Reveals Genetic Diversity and Metabolic Potentials of the Genus *Qipengyuania* and Suggests Fifteen Novel Species

Yang Liu,[a] Tao Pei,[a] Juan Du,[a] Qing Yao,[b] Ming-Rong Deng,[a] Honghui Zhu[a]

[a]Key Laboratory of Agricultural Microbiomics and Precision Application, Ministry of Agriculture and Rural Affairs, State Key Laboratory of Applied Microbiology Southern China, Guangdong Provincial Key Laboratory of Microbial Culture Collection and Application, Guangdong Microbial Culture Collection Center (GDMCC), Institute of Microbiology, Guangdong Academy of Sciences, Guangzhou, People's Republic of China

[b]College of Horticulture, South China Agricultural University, Guangdong Province Key Laboratory of Microbial Signals and Disease Control, Guangzhou, People's Republic of China

**ABSTRACT** Members of the genus *Qipengyuania* are heterotrophic bacteria frequently isolated from marine environments with great application potential in areas such as carotenoid production. However, the genomic diversity, metabolic function, and adaption of this genus remain largely unclear. Here, 16 isolates related to the genus *Qipengyuania* were recovered from coastal samples and their genomes were sequenced. The phylogenetic inference of these isolates and reference type strains of this genus indicated that the 16S rRNA gene was insufficient to distinguish them at the species level; instead, the phylogenomic reconstruction could provide the reliable phylogenetic relationships and confirm 15 new well-supported branches, representing 15 putative novel genospecies corroborated by the digital DNA-DNA hybridization and average nucleotide identity analyses. Comparative genomics revealed that the genus *Qipengyuania* had an open pangenome and possessed multiple conserved genes and pathways related to metabolic functions and environmental adaptation, despite the presence of divergent genomic features and specific metabolic potential. Genetic analysis and pigment detection showed that the members of this genus were identified as carotenoid producers, while some proved to be potentially aerobic anoxygenic photoheterotrophs. Collectively, the first insight into the genetic diversity and metabolic potentials of the genus *Qipengyuania* will contribute to better understanding of the speciation and adaptive evolution in natural environments.

**IMPORTANCE** The deciphering of the phylogenetic diversity and metabolic features of the abundant bacterial taxa is critical for exploring their ecological importance and application potential. *Qipengyuania* is a genus of frequently isolated heterotrophic microorganisms with great industrial application potential. Numerous strains related to the genus *Qipengyuania* have been isolated from diverse environments, but their genomic diversity and metabolic functions remain unclear. Our study revealed a high degree of genetic diversity, metabolic versatility, and environmental adaptation of the genus *Qipengyuania* using comparative genomics. Fifteen novel species of this genus have been established using a polyphasic taxonomic approach, expanding the number of described species to almost double. This study provided an overall view of the genus *Qipengyuania* at the genomic level and will enable us to better uncover its ecological roles and evolutionary history.

**KEYWORDS** *Qipengyuania*, comparative genomics, genetic diversity, metabolic potentials, novel species description.

The genus *Qipengyuania*, belonging to the family *Erythrobacteraceae* (1) of the order *Sphingomonadales* within the class *Alphaproteobacteria*, was first described in 2015 accompanied by the proposal of type species *Qipengyuania sediminis* (2) using a

Address correspondence to Honghui Zhu, zhuhh@gdim.cn.

The authors declare no conflict of interest.

polyphasic taxonomic approach. The genus *Qipengyuania* was initially identified as Gram-negative and chemoheterotrophic bacteria. In August 2020, a new taxonomic framework of the family *Erythrobacteraceae* was established based on the phylogenomic reconstruction of core genes and genomic similarity metrics analyses (3). Based on this taxonomic profile, 10 species of the genus *Erythrobacter* were proposed to be transferred into the genus *Qipengyuania* (3). Later, the species *Erythrobacter flavus* was reclassified into this genus as *Qipengyuania flava* (4). Just recently, a new species, *Qipengyuania soli*, was described along with a proposal for the reclassification of "*Erythrobacter mangrovi*," "*Erythrobacter aureus*," and "*Erythrobacter nanhaiensis*" into the genus *Qipengyuania* (5). The genus *Qipengyuania* currently comprises 13 species with valid names and 3 species with effectively published names (https://lpsn.dsmz.de/genus/qipengyuania). After dissecting these species descriptions, it was found that the genus *Qipengyuania* is physiologically and genetically heterogeneous. However, largely due to insufficient available representative strains and genomes, the genome-based diversity and phylogeny of the genus *Qipengyuania* have not been well studied to date.

Bacteria of the genus *Qipengyuania* have frequently been isolated from various habitats. The type strain of the type species *Q. sediminis* was obtained from a subterrestrial sediment of Qiangtang Basin in Qinghai-Tibetan plateau, People's Republic of China (2); other type strains were isolated from marine environments, such as seawater (6, 7), estuary water (8), intertidal and deep-sea sediments (9, 10), starfish (11), and mangrove soil (5, 12). These results showed that the genus *Qipengyuania* may be worldwide spread in coastal environments, like many other taxa, for example the *Roseobacter* lineage (13). In recent years, the genus *Qipengyuania* has attracted increasing attention due to great potential applications in agriculture, biotechnology, and industry, such as the inhibition of the growth of *Fusarium oxysporum* (14) and some alga causing red tides (15), the production of a halotolerant thermoalkaliphilic esterase (16), sulfur-containing carotenoids (17), and poly-beta-hydroxybutyrate (18), the retardation of the corrosion of 2205 duplex stainless steel in marine conditions by the formation of biofilm (19), and the oxidation of thiosulfate to zero-valent sulfur (20). Several genomic analyses of the genus *Qipengyuania* have provided some vital insights into the psychrophilic adaptations (21) and functions in the biogeochemical cycles of phosphorus (22) and sulfur (20). However, we are still far away from a complete understanding of metabolic commonalities and discrepancies between different species of the genus *Qipengyuania*, due to the lack of genomic comparisons.

To gain a better appreciation for the genomic diversity and metabolic breadth of the genus *Qipengyuania*, 16 isolates were recovered from coastal samples. We then conducted a comparative genomic investigation (pangenome) of the genus *Qipengyuania* to explore its phylogenetic relatedness and industrial application potentials. The results suggest that while the primary metabolic pathways are well conserved (e.g., the central carbohydrate metabolism and ammonium assimilation), certainly predicted metabolisms and genome features were markedly different from what is known of this genus. This study provides insight into some unifying features and ancillary pathways of the genus *Qipengyuania* that may contribute to its survival in marine environments.

## TAXONOMY

**Description of *Qipengyuania xiapuensis* sp. nov.** *Qipengyuania xiapuensis* (xia.-pu.en'sis. N.L. fem. adj. *xiapuensis*, pertaining to Xiapu County, China, from where the type strain was isolated).

Cells are Gram negative, aerobic, rod shaped, and nonmotile. Colonies are circular, slightly convex, smooth, opaque, and yellow with entire margins and about 1 mm in diameter on marine agar (MA) at 28°C for 48 h. Growth at 15 to 40°C (optimum, 28°C), at pH 4.0 to 11.0 (optimum, 7.0), in 0 to 10% (wt/vol) NaCl (optimum, 3%), and on MA, Reasoner's 2A (R2A), tryptic soy agar (TSA), nutrient agar (NA), and LB media. Production of carotenoids. Catalase and oxidase are found to be positive. Hydrolyze Tweens 20, 40, and 60 but not Tween 80, starch, skimmed milk, or carboxymethyl

cellulose. The predominant cellular fatty acids are summed feature 8 and $C_{17:1}$ $\omega 6c$, the major polar lipids are diphosphatidylglycerol, phosphatidylcholine, phosphatidylethanolamine, phosphatidylglycerol, and sphingoglycolipid, and the sole respiratory quinone is ubiquinone-10. The genomic DNA G+C contents are 63.5 to 63.7%.

The type strain, 1NDW9$^T$ (= GDMCC 1.2378$^T$ = KCTC 82662$^T$), was isolated from the aquaculture pond sediment at Xiapu of Ningde of Fujian Province, People's Republic of China. Another strain, 1NDW3 (= GDMCC 1.2377 = KCTC 82609), was also isolated from the same sediment. The GenBank/EMBL/DDBJ accession numbers of 16S rRNA gene and genome sequences of strains 1NDW9$^T$ and 1NDW3 are, respectively, MZ753461 and MZ749491, JAIGNL000000000 and CP081296.

**Description of *Qipengyuania xiamenensis* sp. nov.** *Qipengyuania xiamenensis* (xia.-men.en'sis. N.L. fem. adj. *xiamenensis*, pertaining to Xiamen City, China, from where the type strain was isolated).

Cells are Gram negative, aerobic, rod shaped, and nonmotile. Colonies are circular, slightly convex, smooth, opaque, and yellow with entire margins and about 0.8 mm in diameter on MA at 28°C for 48 h. Growth at 15 to 40°C (optimum, 28°C), at pH 4.0 to 11.0 (optimum, 7.0), in 0 to 10% (wt/vol) NaCl (optimum, 1 to 2%), and on MA, R2A, TSA, NA, and LB media. Production of carotenoids and outer membrane vesicles or their analogues. Catalase and oxidase are found to be positive. Hydrolyze Tweens 20, 40, and 60 but not Tween 80, starch, skimmed milk, or carboxymethyl cellulose. The predominant cellular fatty acids are summed feature 8, $C_{17:1}$ $\omega 6c$, $C_{16:0}$, and summed feature 3, the major polar lipids are diphosphatidylglycerol, phosphatidylcholine, phosphatidylethanolamine, and phosphatidylglycerol, and the sole respiratory quinone is ubiquinone-10. The genomic DNA G+C content is 62.6%.

The type strain, 1XM1-15A$^T$ (= GDMCC 1.2379$^T$ = KCTC 82610$^T$), was isolated from the tidal flat sediment at Xiamen of Fujian Province, People's Republic of China. The GenBank/EMBL/DDBJ accession numbers of 16S rRNA gene and genome sequences of strain 1XM1-15A$^T$ are MZ749493 and JAIGNM000000000, respectively.

**Description of *Qipengyuania gelatinilytica* sp. nov.** *Qipengyuania gelatinilytica* (ge.la.ti.ni.ly'ti.ca. N.L. neut. n. *gelatinum*, gelatin; N.L. masc. adj. *lyticus*, able to dissolve; N.L. fem. adj. *gelatinilytica*, gelatin-dissolving).

Cells are Gram negative, aerobic, rod shaped, and nonmotile. Colonies are circular, slightly convex, smooth, opaque, and yellow with entire margins and about 1 mm in diameter on MA at 28°C for 48 h. Growth at 10 to 40°C (optimum, 28°C), at pH 5.0 to 11.0 (optimum, 7.0), in 2 to 7% (wt/vol) NaCl (optimum, 3%), and on MA, R2A, NA, and LB media; no growth on TSA medium. Production of carotenoids and outer membrane vesicles or their analogues. Catalase and oxidase are found to be positive. Hydrolyze Tweens 20, 40, 60, and 80 but not starch, skimmed milk, or carboxymethyl cellulose. The predominant cellular fatty acids are summed feature 8, $C_{17:1}$ $\omega 6c$, $C_{16:0}$, and summed feature 3, the major polar lipids are diphosphatidylglycerol, phosphatidylcholine, phosphatidylethanolamine, phosphatidylglycerol, and sphingoglycolipid, and the sole respiratory quinone is ubiquinone-10. The genomic DNA G+C content is 62.7%.

The type strain, 1NDH1$^T$ (= GDMCC 1.2372$^T$ = KCTC 82606$^T$), was isolated from the tidal flat sediment at Xiapu of Ningde of Fujian Province, People's Republic of China. The GenBank/EMBL/DDBJ accession numbers of 16S rRNA gene and genome sequences of strain 1NDH1$^T$ are MZ749487 and CP081294, respectively.

**Description of *Qipengyuania vesicularis* sp. nov.** *Qipengyuania vesicularis* (ve.si.c-u.la'ris. N.L. fem. adj. *vesicularis*, pertaining to a small bladder, a vesicle, referring to a vesicle structure associated with the type strain).

Cells are Gram negative, aerobic, rod shaped, and nonmotile. Colonies are circular, slightly convex, smooth, opaque, and yellow with entire margins and about 0.5 mm in diameter on MA at 28°C for 48 h. Growth at 10 to 40°C (optimum, 28°C), at pH 4.0 to 11.0 (optimum, 7.0), in 0.5 to 8% (wt/vol) NaCl (optimum, 3%), and on MA, R2A, NA, TSA, and LB media. Production of carotenoids and outer membrane vesicles or their analogues. Catalase and oxidase are found to be positive. Hydrolyze Tweens 20, 40, and 60 but not Tween 80, starch, skimmed milk, or carboxymethyl cellulose. The

predominant cellular fatty acids are summed feature 8, $C_{16:0}$, and summed feature 3, the major polar lipids are diphosphatidylglycerol, phosphatidylcholine, phosphatidylethanolamine, and phosphatidylglycerol, and the sole respiratory quinone is ubiquinone-10. The genomic DNA G+C content is 62.5%.

The type strain, $1NDH10^T$ (= $GDMCC 1.2373^T$ = $KCTC 82663^T$), was isolated from the tidal flat sediment from Xiapu at Ningde of Fujian Province, People's Republic of China. The GenBank/EMBL/DDBJ accession numbers of 16S rRNA gene and genome sequences of strain $1NDH10^T$ are MZ749488 and JAIGNJ000000000, respectively.

**Description of *Qipengyuania sphaerica* sp. nov.** *Qipengyuania sphaerica* (sphae'r-i.ca. L. fem. adj. *sphaerica*, spherical).

Cells are Gram negative, aerobic, nearly spherical, and nonmotile. Colonies are circular, slightly convex, smooth, opaque, and yellow with entire margins and about 1.2 mm in diameter on MA at 28°C for 48 h. Growth at 10 to 40°C (optimum, 28°C), at pH 4.0 to 11.0 (optimum, 6.0 to 7.0), in 0 to 10% (wt/vol) NaCl (optimum, 2%), and on MA, R2A, NA, TSA, and LB media. Production of carotenoids. Catalase and oxidase are found to be positive. Hydrolyze Tweens 20, 40, and 60 but not Tween 80, starch, skimmed milk, or carboxymethyl cellulose. The predominant cellular fatty acids are summed feature 8 and $C_{17:1}$ $\omega 6c$, the major polar lipids are diphosphatidylglycerol, phosphatidylcholine, phosphatidylethanolamine, phosphatidylglycerol, and sphingoglycolipid, and the sole respiratory quinone is ubiquinone-10. The genomic DNA G+C content is 62.7%.

The type strain, $GH29^T$ (= $GDMCC 1.2371^T$ = $KCTC 82661^T$), was isolated from the tidal flat sediment at Huizhou of Guangdong Province, People's Republic of China. The GenBank/EMBL/DDBJ accession numbers of 16S rRNA gene and genome sequences of strain $GH29^T$ are MZ749498 and JAIGNR000000000, respectively.

**Description of *Qipengyuania aurantiaca* sp. nov.** *Qipengyuania aurantiaca* (au.r-an.ti.a'ca. N.L. fem. adj. *aurantiaca*, orange).

Cells are Gram negative, aerobic, rod shaped, and nonmotile. Colonies are circular, slightly convex, smooth, opaque, and orange with entire margins and about 1 mm in diameter on MA at 28°C for 48 h. Growth at 10 to 40°C (optimum, 28°C), at pH 5.0 to 10.0 (optimum, 6.0 to 7.0), in 0.5 to 7% (wt/vol) NaCl (optimum, 3%), and on MA, R2A, NA, and LB media; no growth on TSA medium. Production of carotenoids and outer membrane vesicles or their analogues. Catalase and oxidase are found to be positive. Hydrolyze Tweens 20, 40, 60, and 80 but not starch, skimmed milk, or carboxymethyl cellulose. The predominant cellular fatty acids are summed feature 8 and summed feature 3, the major polar lipids are diphosphatidylglycerol, phosphatidylcholine, phosphatidylethanolamine, phosphatidylglycerol, and sphingoglycolipid, and the sole respiratory quinone is ubiquinone-10. The genomic DNA G+C content is 63.8%.

The type strain, $1NDH13^T$ (= $GDMCC 1.2375^T$ = $KCTC 82607^T$), was isolated from the tidal flat sediment at Xiapu of Ningde of Fujian Province, People's Republic of China. The GenBank/EMBL/DDBJ accession numbers of 16S rRNA gene and genome sequences of strain $1NDH13^T$ are MZ749489 and CP081295, respectively.

**Description of *Qipengyuania polymorpha* sp. nov.** *Qipengyuania polymorpha* (po.ly.mor'pha. N.L. fem. adj. *polymorpha*, variable in form).

Cells are Gram negative, aerobic, pleomorphic, and nonmotile. Cells are spherical and short rod and/or rod shaped. Colonies are circular, slightly convex, smooth, opaque, and yellow with entire margins and about 0.8 mm in diameter on MA at 28°C for 48 h. Growth at 10 to 40°C (optimum, 28°C), at pH 5.0 to 10.0 (optimum, 6.0 to 8.0), in 2 to 6% (wt/vol) NaCl (optimum, 3%), and on MA, R2A, NA, and LB media; no growth on TSA medium. Production of carotenoids and outer membrane vesicles or their analogues. Catalase and oxidase are found to be positive. Hydrolyze Tween 20 but not Tween 40, 60, or 80, starch, skimmed milk, or carboxymethyl cellulose. The predominant cellular fatty acids are summed feature 8, $C_{17:1}$ $\omega 6c$, and $C_{16:0}$, the major polar lipids are diphosphatidylglycerol, phosphatidylcholine, phosphatidylethanolamine, phosphatidylglycerol, and sphingoglycolipid, and the sole respiratory quinone is ubiquinone-10. The genomic DNA G+C content is 63.3%.

The type strain, $1NDH17^T$ (= $GDMCC 1.2376^T$ = $KCTC 82608^T$), was isolated from the tidal flat sediment of Xiapu County at Ningde City, the Fujian Province, People's Republic

of China. The GenBank/EMBL/DDBJ accession numbers of 16S rRNA gene and genome sequences of strain 1NDH17$^T$ are MZ749490 and JAIGNK000000000, respectively.

**Description of *Qipengyuania aestuarii* sp. nov.** *Qipengyuania aestuarii* (aes.tu.a'ri.i. L. gen. neut. n. *aestuarii*, of a tidal flat).

Cells are Gram negative, aerobic, pleomorphic, and nonmotile. Cells are spherical and short rod and/or rod shaped. Colonies are circular, slightly convex, smooth, opaque, and orange with entire margins and about 1 mm in diameter on MA at 28°C for 48 h. Growth at 10 to 40°C (optimum, 28°C), at pH 4.0 to 11.0 (optimum, 7.0 to 8.0), in 0.5 to 7% (wt/vol) NaCl (optimum, 3%), and on MA, R2A, NA, TSA, and LB media. Production of carotenoids and outer membrane vesicles or their analogues. Catalase and oxidase are found to be positive. Hydrolyze Tweens 20, 40, 60, and 80 but not starch, skimmed milk, or carboxymethyl cellulose. The predominant cellular fatty acids are summed feature 8, summed feature 3, $C_{14:0}$ 2OH, and $C_{18:1}$ $\omega 7c$ 11-methyl, the major polar lipids are diphosphatidylglycerol, phosphatidylcholine, phosphatidylethanolamine, phosphatidylglycerol, and sphingoglycolipid, and the sole respiratory quinone is ubiquinone-10. The genomic DNA G+C content is 61.1%.

The type strain, GH1$^T$ (= GDMCC 1.2370$^T$ = KCTC 82605$^T$), was isolated from the tidal flat sediment at Huizhou of Guangdong Province, People's Republic of China. The GenBank/EMBL/DDBJ accession numbers of 16S rRNA gene and genome sequences of strain GH1$^T$ are MZ749496 and JAIGNP000000000, respectively.

**Description of *Qipengyuania huizhouensis* sp. nov.** *Qipengyuania huizhouensis* (hui.zhou.en'sis. N.L. fem. adj. *huizhouensis*, pertaining to Huizhou, People's Republic of China, from where the type strain was isolated).

Cells are Gram negative, aerobic, rod shaped, and nonmotile. Colonies are circular, slightly convex, smooth, opaque, and yellow with entire margins and about 0.8 mm in diameter on MA at 28°C for 48 h. Growth at 10 to 40°C (optimum, 28°C), at pH 4.0 to 11.0 (optimum, 7.0), in 0 to 10% (wt/vol) NaCl (optimum, 2 to 3%), and on MA, R2A, NA, TSA, and LB media. Production of carotenoids and outer membrane vesicles or their analogues. Catalase and oxidase are found to be positive. Hydrolyze Tweens 20, 40, and 60 but not Tween 80, starch, skimmed milk, or carboxymethyl cellulose. The predominant cellular fatty acids are summed feature 8, $C_{15:0}$ 2OH, and $C_{17:1}$ $\omega 6c$, the major polar lipids are diphosphatidylglycerol, phosphatidylcholine, phosphatidylethanolamine, phosphatidylglycerol, and sphingoglycolipid, and the sole respiratory quinone is ubiquinone-10. The genomic DNA G+C content is 60.6%.

The type strain, YG19$^T$ (= GDMCC 1.2369$^T$ = KCTC 82604$^T$), was isolated from the tidal flat sediment at Huizhou of Guangdong Province, People's Republic of China. The GenBank/EMBL/DDBJ accession numbers of 16S rRNA gene and genome sequences of strain YG19$^T$ are MZ749500 and JAIGNT000000000, respectively.

**Description of *Qipengyuania psychrotolerans* sp. nov.** *Qipengyuania psychrotolerans* (psy.chro.to'le.rans. Gr. masc. adj. *psychros*, cold; L. pres. part. *tolerans*, tolerating; N.L. part. adj. *psychrotolerans*, tolerating cold temperature).

Cells are Gram negative, aerobic, rod shaped, and nonmotile. Colonies are circular, slightly convex, smooth, opaque, and orange with entire margins and about 0.8 mm in diameter on MA at 28°C for 48 h. Growth at 4 to 37°C (optimum, 28°C), at pH 5.0 to 10.0 (optimum, 7.0), in 2 to 7% (wt/vol) NaCl (optimum, 3%), and on MA, R2A, NA, TSA, and LB media. Production of carotenoids. Catalase and oxidase are found to be positive. Hydrolyze Tween 20 but not Tween 40, 60, or 80, starch, skimmed milk, or carboxymethyl cellulose. The predominant cellular fatty acids are summed feature 8, summed feature 3, and $C_{17:1}$ $\omega 6c$, the major polar lipids are diphosphatidylglycerol, phosphatidylcholine, phosphatidylethanolamine, phosphatidylglycerol, and sphingoglycolipid, and the sole respiratory quinone is ubiquinone-10. The genomic DNA G+C content is 60.1%.

The type strain, 1XM2-8$^T$ (= GDMCC 1.2380$^T$ = KCTC 82611$^T$), was isolated from the tidal flat sediment at Xianmen of Fujian Province, People's Republic of China. The GenBank/EMBL/DDBJ accession numbers of 16S rRNA gene and genome sequences of strain 1XM2-8$^T$ are MZ749492 and CP081297, respectively.

**Description of *Qipengyuania intermedia* sp. nov.** *Qipengyuania intermedia* (in.ter.-me'di.a. L. fem. adj. *intermedia*, that is between, intermediate).

Cells are Gram negative, aerobic, rod shaped, and nonmotile. Colonies are circular, slightly convex, smooth, opaque, and orange with entire margins and about 0.8 mm in diameter on MA at 28°C for 48 h. Growth at 15 to 42°C (optimum, 28°C), at pH 5.0 to 11.0 (optimum, 7.0 to 8.0), in 0.5 to 7% (wt/vol) NaCl (optimum, 3%), and on MA, R2A, NA, and LB media; no growth on TSA medium. Production of carotenoids. Catalase and oxidase are found to be positive. Hydrolyze Tweens 20, 40, and 60 but not Tween 80, starch, skimmed milk, or carboxymethyl cellulose. The predominant cellular fatty acids are summed feature 8, summed feature 3, $C_{17:1}$ $\omega6c$, and $C_{16:0}$, the major polar lipids are diphosphatidylglycerol, phosphatidylcholine, phosphatidylethanolamine, phosphatidylglycerol, and sphingoglycolipid, and the sole respiratory quinone is ubiquinone-10. The genomic DNA G+C content is 62.5%.

The type strain, GH38$^T$ (= GDMCC 1.2368$^T$ = KCTC 82603$^T$), was isolated from the tidal flat sediment at Huizhou of Guangdong Province, People's Republic of China. The GenBank/EMBL/DDBJ accession numbers of 16S rRNA gene and genome sequences of strain GH38$^T$ are MZ749499 and JAIGNS000000000, respectively.

**Description of *Qipengyuania proteolytica* sp. nov.** *Qipengyuania proteolytica* (pro.-te.o.ly'ti.ca. N.L. neut. n. *proteinum*, protein; N.L. fem. adj. *lytica*, able to loosen, able to dissolve; from Gr. masc. adj. *lytikos*, dissolving; N.L. fem. adj. *proteolytica*, protein dissolving)

Cells are Gram negative, aerobic, short rod shaped, and nonmotile. Colonies are circular, slightly convex, smooth, opaque, and yellow with entire margins and about 1 mm in diameter on MA at 28°C for 48 h. Growth at 10 to 40°C (optimum, 28°C), at pH 5.0 to 10.0 (optimum, 7.0), in 2 to 9% (wt/vol) NaCl (optimum, 3%), and on MA, R2A, NA, and LB media; no growth on TSA medium. Production of carotenoids. Catalase and oxidase are found to be positive. Hydrolyze Tween 20 and skimmed milk but not Tween 40, 60, or 80, starch, or carboxymethyl cellulose. The predominant cellular fatty acids are summed feature 8, summed feature 3, and $C_{17:1}$ $\omega6c$, the major polar lipids are diphosphatidylglycerol, phosphatidylcholine, phosphatidylethanolamine, phosphatidylglycerol, and sphingoglycolipid, and the sole respiratory quinone is ubiquinone-10. The genomic DNA G+C content is 65.0%.

The type strain, 6B39$^T$ (= GDMCC 1.2364$^T$ = KCTC 82599$^T$), was isolated from the mangrove soil at Qi'ao Island of Guangdong Province, People's Republic of China. The GenBank/EMBL/DDBJ accession numbers of 16S rRNA gene and genome sequences of strain 6B39$^T$ are MZ749495 and JAIGNN000000000, respectively.

**Description of *Qipengyuania aerophila* sp. nov.** *Qipengyuania aerophila* (ae.ro'-phi.la. Gr. masc. n. *aer*, air; Gr. masc. adj. *philos*, loving; N.L. fem. adj. *aerophila*, air loving)

Cells are Gram negative, aerobic, rod shaped, and motile by a lateral flagellum. Colonies are circular, slightly convex, smooth, opaque, and yellow with entire margins and about 1.6 mm in diameter on MA at 28°C for 48 h. Growth at 10 to 40°C (optimum, 28°C), at pH 4.0 to 11.0 (optimum, 7.0 to 8.0), in 0.5 to 7% (wt/vol) NaCl (optimum, 3%), and on MA, R2A, NA, TSA, and LB media. Production of carotenoids and outer membrane vesicles or their analogues. Catalase and oxidase are found to be positive. Hydrolyze Tweens 20, 40, 60, and 80 but not starch, skimmed milk, or carboxymethyl cellulose. The predominant cellular fatty acids are summed feature 8, summed feature 3, $C_{16:0}$ 2OH, and $C_{14:0}$ 2OH, the major polar lipids are diphosphatidylglycerol, phosphatidylcholine, phosphatidylethanolamine, phosphatidylglycerol, and sphingoglycolipid, and the sole respiratory quinone is ubiquinone-10. The genomic DNA G+C content is 60.7%.

The type strain, GH25$^T$ (= GDMCC 1.2366$^T$ = KCTC 82601$^T$), was isolated from the tidal flat sediment at Huizhou of Guangdong Province, People's Republic of China. The GenBank/EMBL/DDBJ accession numbers of 16S rRNA gene and genome sequences of strain GH25$^T$ are MZ749497 and JAIGNQ000000000, respectively.

**Description of *Qipengyuania qiaonensis* sp. nov.** *Qipengyuania qiaonensis* (qi.ao.-nen'sis. N.L. fem. adj. *qiaonensis*, pertaining to Qi'ao Island, People's Republic of China, from where the type strain was isolated).

Cells are Gram negative, aerobic, rod shaped, and motile by a polar flagellum. Colonies are circular, slightly convex, smooth, opaque, and yellow with entire margins and about

0.8 mm in diameter on MA at 28°C for 48 h. Growth at 15 to 40°C (optimum, 28°C), at pH 4.0 to 10.0 (optimum, 7.0), in 0 to 9% (wt/vol) NaCl (optimum, 3%), and on MA, R2A, NA, TSA, and LB media. Production of carotenoids. Catalase and oxidase are found to be positive. Hydrolyze Tweens 20, 40, and 60 but not Tween 80, starch, skimmed milk, or carboxymethyl cellulose. The predominant cellular fatty acids are summed feature 8, summed feature 3, and $C_{16:0}$, the major polar lipids are diphosphatidylglycerol, phosphatidylcholine, phosphatidylethanolamine, phosphatidylglycerol, and sphingoglycolipid, and the sole respiratory quinone is ubiquinone-10. The genomic DNA G+C content is 62.0%.

The type strain, 6D47A$^T$ (= GDMCC 1.2365$^T$ = KCTC 82600$^T$), was isolated from the mangrove soil at Qi'ao Island of Guangdong Province, People's Republic of China. The GenBank/EMBL/DDBJ accession numbers of 16S rRNA gene and genome sequences of strain 6D47A$^T$ are MZ749494 and JAIGNO000000000, respectively.

**Description of *Qipengyuania mesophila* sp. nov.** *Qipengyuania mesophila* (me.so'-phi.la. Gr. masc. adj. *mesos*, medium; Gr. masc. adj. *philos*, loving; N.L. fem. adj. *mesophila*, medium temperature loving, mesophilic).

Cells are Gram negative, aerobic, rod shaped, and motile by a polar flagellum. Colonies are circular, slightly convex, smooth, opaque, and yellow with entire margins and about 0.6 mm in diameter on MA at 28°C for 48 h. Growth at 15 to 40°C (optimum, 28°C), at pH 5.0 to 11.0 (optimum, 7.0), in 0 to 10% (wt/vol) NaCl (optimum, 3%), and on MA, R2A, NA, TSA, and LB media. Production of carotenoids. Catalase and oxidase are found to be positive. Hydrolyze Tweens 20, 40, and 60 but not Tween 80, starch, skimmed milk, or carboxymethyl cellulose. The predominant cellular fatty acids are summed feature 8, summed feature 3, and $C_{17:1}$ $\omega6c$, the major polar lipids are diphosphatidylglycerol, phosphatidylcholine, phosphatidylethanolamine, phosphatidylglycerol, and sphingoglycolipid, and the sole respiratory quinone is ubiquinone-10. The genomic DNA G+C content is 65.0%.

The type strain, YG27$^T$ (= GDMCC 1.2367$^T$ = KCTC 82602$^T$), was isolated from the tidal flat sediment at Huizhou of Guangdong Province, People's Republic of China. The GenBank/EMBL/DDBJ accession numbers of 16S rRNA gene and genome sequences of strain YG27$^T$ are MZ749486 and JAIGNU000000000, respectively.

## RESULTS AND DISCUSSION

***Qipengyuania* often occurs in coastal areas.** Before starting with this research, to investigate the diversity and biogeography of the aerobic anoxygenic phototrophic bacteria in coastal regions in the People's Republic of China, more than 1,000 isolates were recovered from diverse coastal samples (most isolates unpublished). Among these isolates, only 16 isolates related to the genus *Qipengyuania* were used in this study. As shown in Fig. S1a and Table S2, the two isolates 1NDW9$^T$ and 1NDW3 and the four isolates 1NDH1$^T$, 1NDH10$^T$, 1NDH13$^T$, and 1NDH17$^T$ were isolated from an aquaculture pond sediment and a tidal flat sediment, respectively, the two isolates 1XM1-15A$^T$ and 1XM2-8$^T$ were obtained from the two tidal flat sediments, the six isolates GH29$^T$, GH1$^T$, YG19$^T$, GH38$^T$, GH25$^T$, and YG27$^T$ were acquired from tidal flat sediments, and the two isolates 6B39$^T$ and 6D47A$^T$ were from mangrove soil. Combined with isolated sources of these isolates and all type strains (Fig. S1b and Table S2), the genus *Qipengyuania* seems to occur frequently in coastal areas.

**The 16S rRNA gene cannot accurately differentiate the *Qipengyuania* species.** As a first step toward identifying 16 isolates, almost complete 16S rRNA gene sequences were obtained by the Sanger sequencing. Given that 16S rRNA gene similarities between isolates and reference type strains were above the 94.5% threshold for genus delineation (23), these isolates were classified into the genus *Qipengyuania*. The ML phylogenetic tree based on 16S rRNA gene sequences showed that these isolates clustered with all type strains of the genus *Qipengyuania* (Fig. 1a), showing their phylogenetically close relationships. However, almost all similarities (a mean of 97.7% and a median of 97.9%) of 16S rRNA gene sequences among 31 strains were above the 97% threshold for species delineation (Fig. S1c), implying that the 16S rRNA gene cannot accurately differentiate closely related species of this genus. Low bootstrap values at

**(a)** 16S rRNA gene-based tree

**(b)** 1146 core genes-based tree

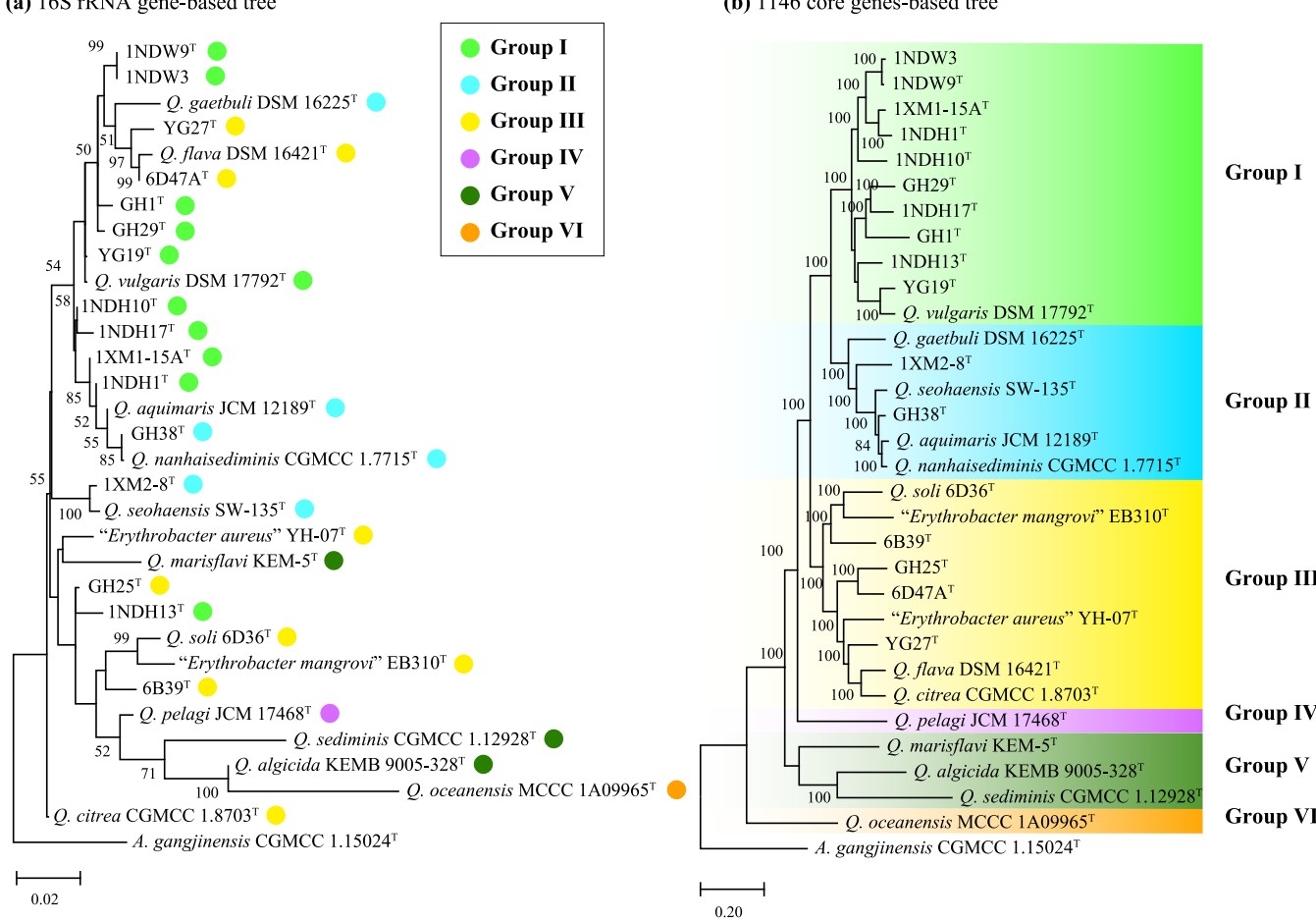

**FIG 1** Phylogenetic trees of bacteria within the genus *Qipengyuania* respectively based on 16S rRNA gene sequences (a) and the *Qipengyuania*-specific core genes (b). The two trees are inferred using the maximum likelihood method. Type strain *A. gangjinensis* CGMCC 1.15024 (accession numbers EU428782 and CP018097) is used as an outgroup. The species names are effectively but not yet validly published and thus are in quotation marks. Bootstrap values greater than 50/80% are shown at branch points. Bar, 0.02/0.2 represents the number of substitutions per site.

many inner nodes of the 16S rRNA gene-based tree also mirrored its weak robustness and unstable topologies. These results demonstrated that the 16S rRNA gene can be used to classify these isolates into the genus *Qipengyuania* but cannot be used to infer their accurate phylogenetic relationships at the species level.

**Genomic features of *Qipengyuania* reflect a high degree of genetic diversity.** The 15 genomes of type strains within the genus *Qipengyuania* are publicly available. By adding 16 genomes of new isolates, a total of 31 genomes were used in this study. Based on the estimation by CheckM, all genomes were to be considered high quality based on more than 98.5% of completeness and less than 1% of contamination (Table S1). The general genomic characteristics for all strains are summarized in Table S1. More specifically, the genomic sizes of these strains showed a wide range from 2.42 Mbp for *Q. sediminis* CGMCC 1.12928$^T$ to 3.39 Mbp for strain 6D47A$^T$, giving an overall mean of 2.89 $\pm$ 0.22 Mbp, which is smaller than most other genera of the family *Erythrobacteraceae* (Fig. S2a). The *Qipengyuania* strains' genomic DNA G+C contents ranged from 60.1% for strain 1XM2-8$^T$ to 66.8% for *Q. sediminis* CGMCC 1.12928$^T$, with an average of 62.8 $\pm$ 1.5%, which were in the range of 52.8 to 68.2% in the family *Erythrobacteraceae* (Fig. S2b). The 27 high-quality draft genomes were detected with an average of nine scaffolds per genome corresponding to an average $N_{50}$ score of 1.50 Mbp (Table S1). The 2,378 to 3,377 genes and 2,326 to 3,325 coding DNA sequences were identified in these genomes. The number of predicted tRNA of genomes varied from 44 to 53. Except for *Q. citrea* CGMCC 1.8703$^T$, one or two rRNA operons in

each genome were found. These differences in general genomic characteristics can, to a certain extent, reflect a high degree of genetic diversity of the genus *Qipengyuania*.

**Phylogenomics provides robust evolutionary relationships of *Qipengyuania*.** One goal of this study was to establish the reliable phylogenetic relationships of the *Qipengyuania* species using two phylogenies, derived from two different sets of genes, the *Qipengyuania*-specific core genes (QCG) and bacterial core genes (BCG); the gene numbers of the two sets were 1,146 from the pangenome analysis below and 92 from the previous report (24), respectively. Despite the differences in the number of informative sites, both QCG and BCG phylogenies provided concordant and reliable phylogenetic relationships among 16 isolates, supported by high bootstrap values on almost all nodes in two phylogenetic trees (Fig. 1b and Fig. S3). In both QCG and BCG phylogenies, six distinct monophyletic clades were identified and tentatively designated groups I, II, III, IV, V, and VI (Fig. 1b). Interestingly, all 10 isolates clustered into group I except *Q. vulgaris* DSM 17792$^T$. Group II as the sister taxon of group I contained two isolates and four type strains. Group III included four isolates and five type strains. The remaining three clades (groups IV, V, and VI) located at the base of the phylogenomic trees were phylogenetically distant from groups I, II, and III.

In both phylogenomic trees, the phylogenetic relationships of most type strains in this study were consistent with those from previous reports (3, 5), but the two species "*Erythrobacter aureus*" and "*Erythrobacter mangrovi*" fell into group III and thus were reclassified into the genus *Qipengyuania*, in agreement with the recent study (5). Considering that the names of the two species were effectively published but not validly published, they should be formally reclassified into the genus *Qipengyuania* if they get to be included in a validation list in the *International Journal of Systematic and Evolutionary Microbiology*. Among 16 isolates, based on the genetic distances of the genome-based trees, strain 1NDW3 was closely related to strain 1NDW9$^T$; the remaining isolates formed 14 distinctly separate branches. The phylogenomic analysis indicated that these isolates present clear and independent phylogenetic positions distinct from all known species of the genus *Qipengyuania*, and the two published species above await to be reclassified in the future.

**OGRI supports 15 novel genospecies of the genus *Qipengyuania*.** The phylogenomic analysis can provide an important depiction of phylogenetic relationships of different strains within the genus *Qipengyuania* but does not translate directly into the overall genome similarity. Therefore, the overall genome relatedness indices (OGRI) including DNA-DNA hybridization (dDDH) and average nucleotide identity (ANI) were used to evaluate genomic similarity. Species determination has been recognized as the cornerstone of microbial diversity analysis and functional comparison. In this study, the species assignments of 16 isolates were first determined using the dDDH values. As shown in the lower left of Fig. 2a, 78.8% dDDH between the two isolates 1NDW3 and 1NDW9$^T$ was slightly above the recognized species separation threshold of 70% (25), indicating that the isolates should belong to the same species. The dDDH values of each of the 2 strains and congeneric strains including 14 other isolates and reference type strains were well below this threshold, demonstrating that the 2 isolates should represent a novel genospecies. Likewise, the dDDH values between 14 isolates and reference type strains ranged from 18.2 to 47.3% (Fig. 2a), which is far lower than the cutoff value for species demarcation. As a result, the analysis of the dDDH values demonstrated that 16 isolates should be 15 novel genospecies of the genus *Qipengyuania*.

The genomic similarities between 16 isolates and reference type strains were also characterized, as shown in the upper right of Fig. 2a. The ANI analysis indicated that isolates 1NDW3 and 1NDW9$^T$ shared a 97.7% ANI value and 18.2 to 26.8% ANI values with 14 other isolates and reference type strains, indicating that they should be conspecific and a novel genospecies by comparing the ANI cutoff value of 95 to 96% for delineating species (26). The ANI values between the remaining 14 isolates and reference type strains ranged from 77.7 to 92.8%, which is below the ANI cutoff value for species delineation but above the ANI genus demarcation boundary of 73.98% (27).

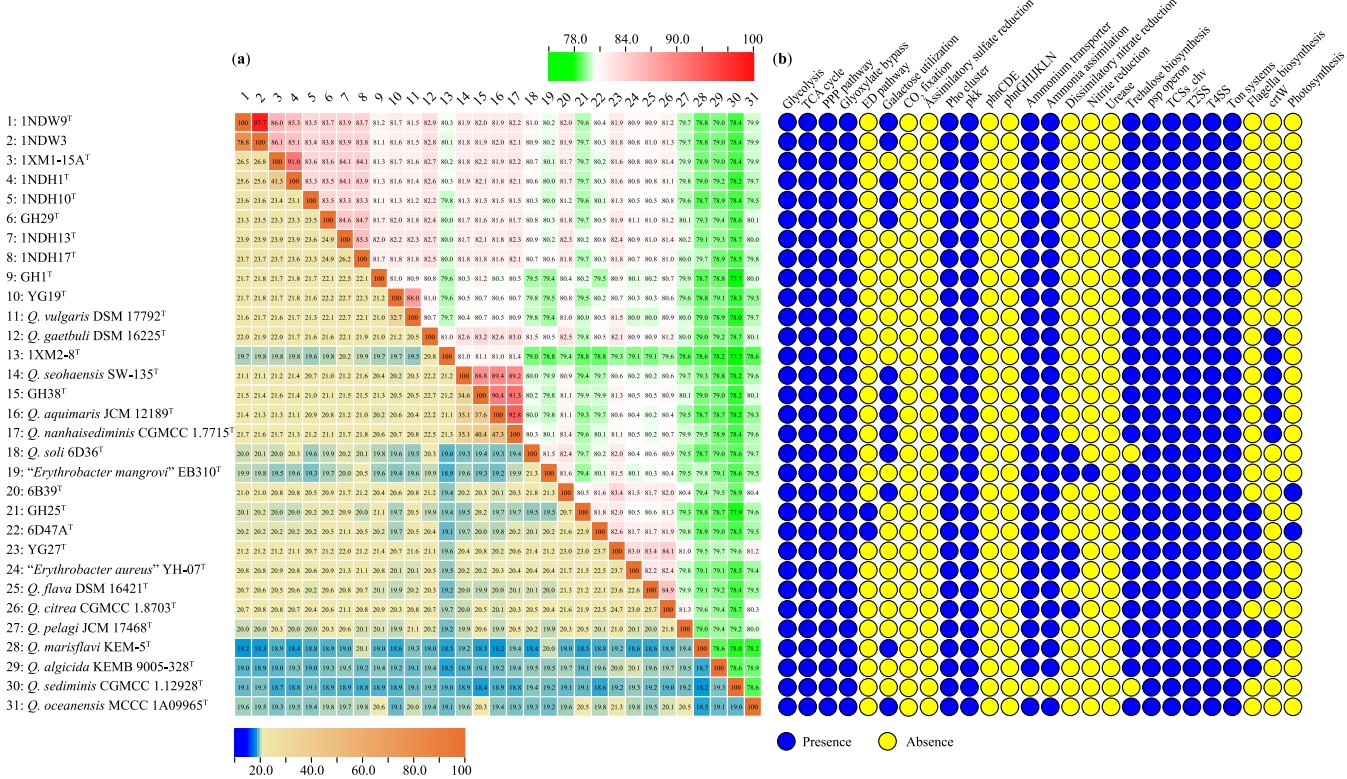

**FIG 2** The heatmap of dDDH values at the lower left and ANI values at the upper right (a), and the distribution of key metabolic genes and pathways among strains of the genus *Qipengyuania* (b). The dDDH and ANI values are visualized using the "Heatmap" tool of the TBtools. The key metabolic genes and pathways are annotated by RAST.

The consistent results from the dDDH and ANI analyses revealed that 16 isolates taxonomically represent 15 putative novel genospecies of the genus *Qipengyuania*.

**The genus *Qipengyuania* possesses an open pangenome.** The pangenome defines the entire genomic repertoire for a given phylogenetic taxon and decodes its possible lifestyles by utilizing predicted protein sequences (28). While the presence of a particular pathway does not definitively prescribe function, whole-genome comparisons can reveal evolutionary conservation or divergence and provide valuable information about the functional potential in each genome. In this study, we identified the common set of gene families (core genome essential for microbial basic lifestyle), the unique set of gene families (unique genome related to the adaptation and some unique characteristics), and the ancillary set of gene families, which were found in at least two, but not all, genomes (accessory genome).

A total of 15,172 gene families were identified based on the pangenome analysis. The numbers of core genes, accessory genes, and unique genes were 1,146 (7.6%), 5,283 (34.8%), and 8,743 (57.6%), respectively (Fig. 3a). The percentages of core genes, accessory genes, and unique genes in each genome varied considerably (Table S3), probably indicating a heterogeneous pattern of genomic diversity and evolution. The curves of pangenome and core genome sizes indicated an open pangenome of the genus *Qipengyuania*, which was supported by the parameter b value (0.651913) in the power-law regression function (Fig. 3b). That is, new unique genes would further increase the pangene pool size with the increase of the genome number (Fig. S4). In general, an open pangenome is predominant in bacteria that are susceptible to horizontal gene transfer (HGT) (29). Based on the power-law regression model used in this study, similar results were determined in the species *Enterococcus faecalis* (30) and *Acinetobacter baumannii* (31) and the genera *Metallosphaera* (32) and *Corynebacterium* (33).

**The genus *Qipengyuania* shares unifying functional categories.** Functional characterization from core, accessory, and unique genes was conducted using the Clusters of

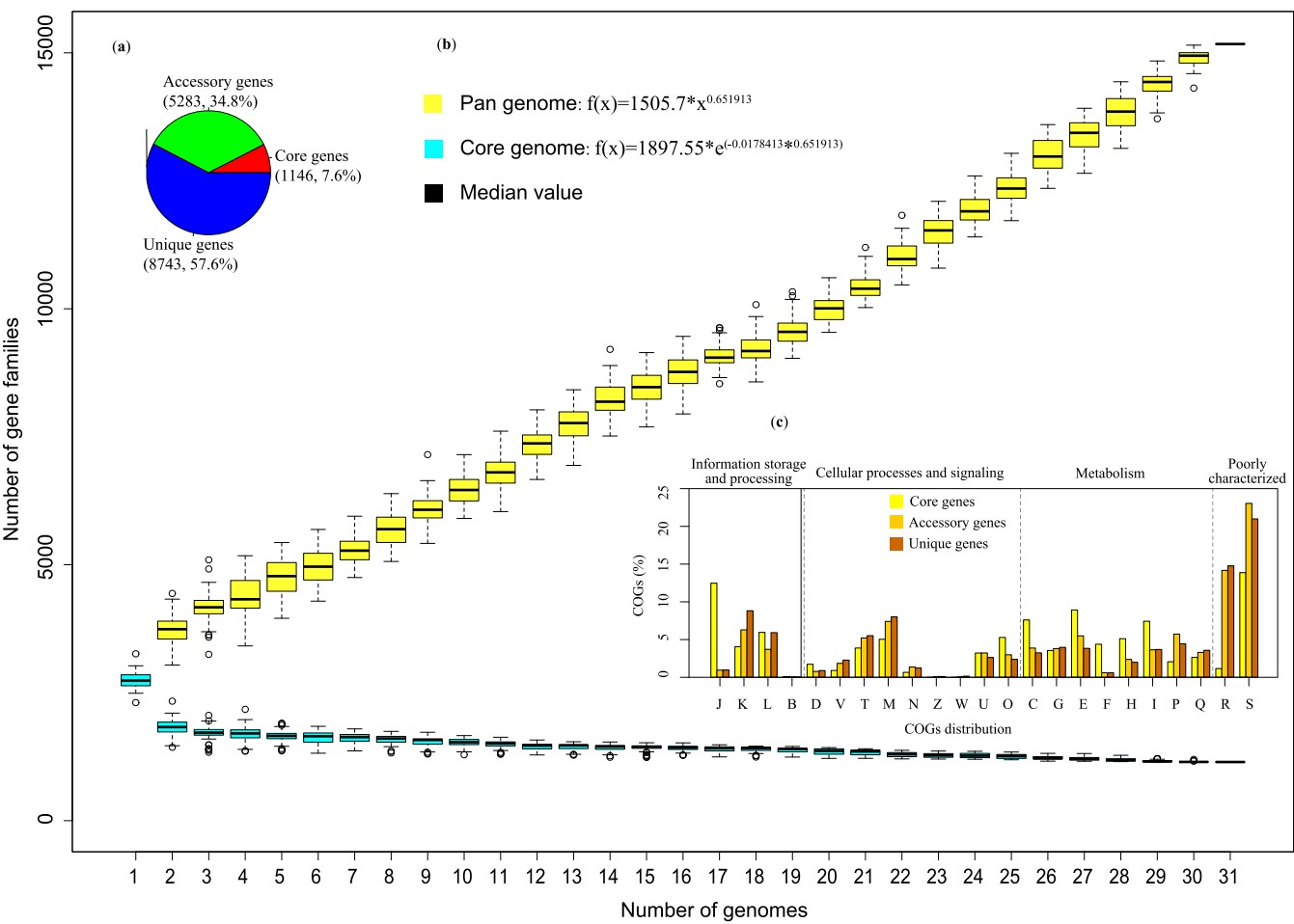

**FIG 3** The pangenome analysis of bacteria within the genus *Qipengyuania*. (a) The numbers and proportions of core genes, accessory genes, and unique genes. (b) The boxplots of the pangenome (yellow) and core genome (cyan) of the 31 analyzed genomes. (c) The proportions of COGs functional categories of core genes, accessory genes, and unique genes. J, translation, ribosomal structure and biogenesis; K, transcription; L, replication, recombination, and repair; B, chromatin structure and dynamics; D, cell cycle control, cell division, and chromosome partitioning; V, defense mechanisms; T, signal transduction mechanisms; M, cell wall/membrane/envelope biogenesis; N, cell motility; Z, cytoskeleton; W, extracellular structures; U, intracellular trafficking, secretion, and vesicular transport; O, posttranslational modification, protein turnover, and chaperones; C, energy production and conversion; G, carbohydrate transport and metabolism; E, amino acid transport and metabolism; F, nucleotide transport and metabolism; H, coenzyme transport and metabolism; I, lipid transport and metabolism; P, inorganic ion transport and metabolism; Q, secondary metabolite biosynthesis, transport, and catabolism; R, general function prediction only; S, function unknown.

Orthologous Groups (COG) annotations. The distribution of COG categories was not homogeneous in core, accessory, or unique genes (Table S4). As shown in Fig. 3c, core genes were assigned into many important and central biological functional categories against accessory genes and unique genes, such as translation, ribosomal structure, and biogenesis (category J, the same below), amino acid transport and metabolism (E), energy production and conversion (C), and lipid transport and metabolism (I). In contrast, the highest percentage of accessory genes was related to inorganic ion transport and metabolism (P). Unique genes were categorized mainly into transcription (K), cell wall/membrane/envelope biogenesis (M), and signal transduction mechanisms (T). A few core genes (182) and many accessory genes and unique genes (1,728 and 1,909) were assigned to COG categories R and S, suggesting that their functions await to be studied further. Core, accessory, and unique genes had nearly equal distribution in several COG categories, such as intracellular trafficking, secretion, and vesicular transport (U) and carbohydrate transport and metabolism (G). The genomic diversity and specificity of these strains within the genus *Qipengyuania* may reflect the distinct survival strategies in different environments to some extent.

**The genus *Qipengyuania* harbors expansive metabolic potentials and adaptations.
(i) Conserved central carbon metabolism.** Carbohydrate metabolism is a fundamental biochemical process that ensures constant supplies of carbon and energy to living

cells. The central carbohydrate metabolism pathways, including glycolysis, the tricarboxylic acid cycle, the pentose phosphate pathway, and the glyoxylate bypass, were conserved in all strains of the genus *Qipengyuania* (Fig. 2b). The glyoxylate bypass is an important acclimation strategy for marine heterotrophic bacteria when subjected to Fe-limitation (34), which may also be true in the genus *Qipengyuania*. Only two strains, GH25[T] and 6D47A[T], contained the Entner-Doudoroff pathway (Fig. 2b), which may make them more resistant to oxidative stress from glucose catabolism (35). The genes encoding galactokinase and galactose-1-phosphate uridylyltransferase were found in 13 strains, indicating that galactose could be utilized as a carbon source (Fig. 2b).

**(ii) Low-abundance genes encoding CAZymes.** Carbohydrate active enzymes (CAZymes) are associated with the biosynthesis, binding, and catabolism of carbohydrates (36). The genus *Qipengyuania* contained some genes encoding glycosyltransferases (GTs) and glycoside hydrolases and a few genes encoding carbohydrate esterases; those encoding GTs were the most abundant (Fig. S5). Strains *Q. algicida* KEMB 9005-328[T] and 6D47A[T] possessed the most and second-most genes encoding CAZymes, with 35 and 32, which were almost three times as many as strain *Q. sediminis* CGMCC 1.12928[T], which possessed the fewest genes, with 12. Contrary to the genera *Arthrobacter* (37) and *Bacillus* (38), the low number of genes encoding CAZymes may mirror the weak ability of the genus *Qipengyuania* to utilize different carbohydrates as carbon and energy sources, which was confirmed by the carbohydrate utilization tests described below.

**(iii) *Qipengyuania* may prefer ammonium as nitrogen source**. Nitrogen is an essential nutrient and often limiting for microorganisms in all natural ecosystems, especially in marine environments (39). As such, microorganisms have evolved a variety of strategies to acquire nitrogen from the surroundings. Exception for strain *Q. sediminis* CGMCC 1.12928[T], all investigated strains of the genus *Qipengyuania* harbored homologues for ammonia uptake (Fig. 2b), including the *amtB* gene encoding ammonium transporter, the *gdhA/gdh2* gene encoding glutamate dehydrogenase, and the *gltABD* genes encoding glutamine synthetase. It was reported that the *glnK* and *amtB* genes could interact directly via protein-protein interaction to regulate ammonium assimilation during nitrogen starvation (40). Unlike the ammonium assimilation, the gene cluster *narIJHGKR* encoding enzymes responsible for the dissimilatory nitrate reduction existed in only five strains, including strains YG27[T], *Q. soli* 6D36[T], "*Erythrobacter mangrovi*" EB310[T], "*Erythrobacter aureus*" YH-07[T], and *Q. citrea* CGMCC 1.8703[T] (Fig. 2b). Unexpectedly, the genes encoding nitrite reductase were found only in strain "*Erythrobacter mangrovi*" EB310[T]. These analyses suggested that bacteria of the genus *Qipengyuania* may prefer ammonium as a nitrogen source, although a few bacteria could utilize nitrate (41) and nitrite (12).

**(iv) *Qipengyuania* holds the potential to utilize organosulfur compounds and inorganic phosphates as sulfur and phosphorus sources**. Seawater and marine sediments contain a high concentration of sulfate (42). However, the genes related to assimilatory sulfate reduction were not detected in members of the genus *Qipengyuania* (Fig. 2b), indicating that they should not utilize inorganic sulfur. Likewise, all strains lacked the genes encoding the transporter responsible for the alkanesulfonate acquisition. Therefore, a reasonable inference was that the genus *Qipengyuania* may utilize other sulfur-containing organic compounds, such as amino acids, as sulfur sources. All strains possessed a high-affinity phosphate acquisition system (PstSCAB) and a regulatory system (PhoUBR) (Fig. 2b) that mediate inorganic phosphate transmembrane transport and regulation to ensure the sufficient supply under phosphate-limiting conditions (43). Consistent with the known study (21), all strains harbored genes encoding polyphosphate kinase responsible for the synthesis and utilization of inorganic polyphosphate (Fig. 3b), while they did not contain genes related to organic phosphate transport (PhnCDE) and cleavage (PhnGHIJKLN). These features indicated that the genus *Qipengyuania* may prefer to utilize inorganic phosphate over organic phosphate.

**(v) *Qipengyuania* evolves diverse adaptation mechanisms**. Because almost all strains of the genus *Qipengyuania* were isolated from marine habitats, a genomic scan for the underlying adaptation to heterogeneous and variable marine environments

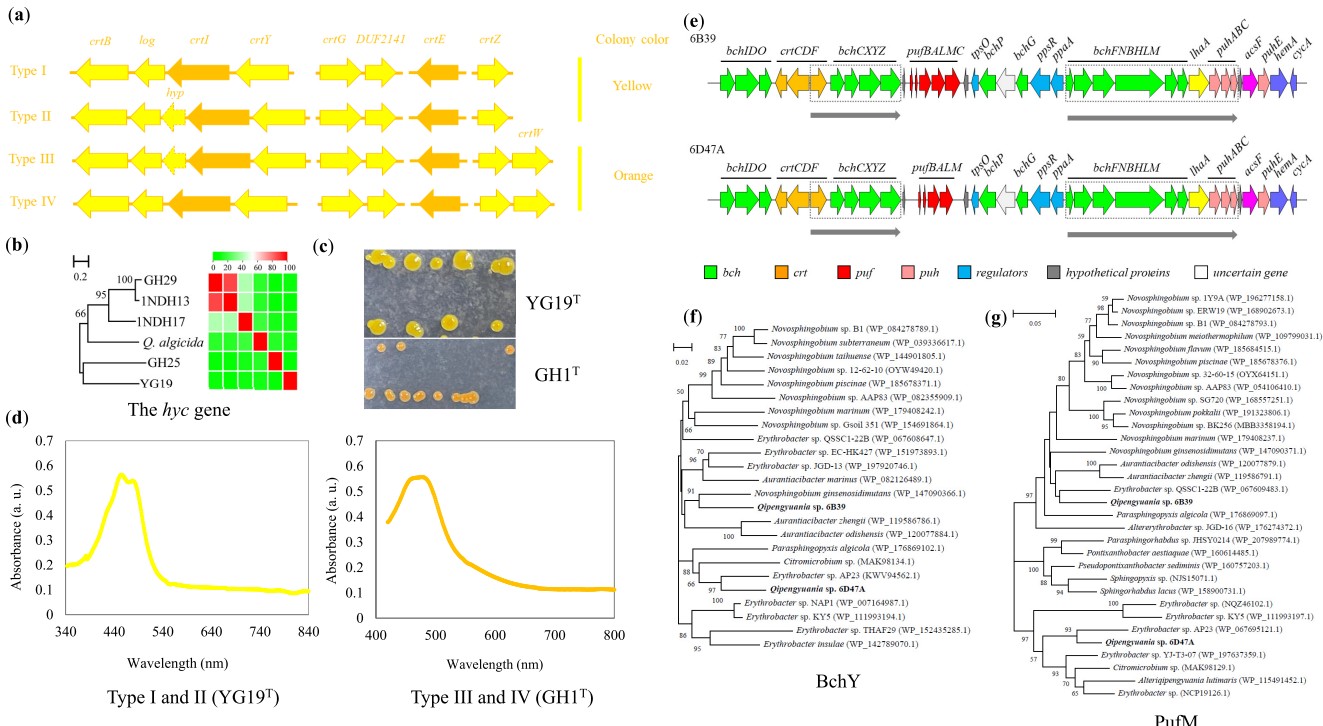

**FIG 4** The genes and pathways related to carotenoid and bacteriochlorophyll *a* biosynthesis of bacteria within the genus *Qipengyuania*. (a) The arrangement of the carotenoid biosynthesis-related genes. (b) The NJ phylogenetic tree based on the *hyc* gene sequences and the heatmap of pairwise similarities. Bootstrap values greater than 50% are shown at branch points. Bar, 0.2 represents nucleotide substitutions per site. (c) Colony colors of two representative strains, YG19$^T$ and GH1$^T$. (d) The UV-VIS absorption spectra of carotenoids of two representative strains, YG19$^T$ and GH1$^T$. (e) The arrangement of the bacteriochlorophyll *a* biosynthesis-related genes. (f and g) The phylogenetic trees based on BchY and PufM sequences. The sequences of reference strains are selected by a concise Microbial Protein BLAST search. Accession numbers of individuals are denoted in parentheses. Bootstrap values greater than 50% are shown at branch points. Bar, 0.02/0.05 nucleotide substitution rate (*K*nuc) units.

was investigated in this study. Except for type strain *Q. soli* 6D36$^T$, a pathway responsible for the trehalose biosynthesis was detected in each strain based on the presence of genes encoding a trehalose-6-phosphate phosphatase and an alpha-trehalose-phosphate synthase, respectively (Fig. 2b). The trehalose accumulation can contribute to coping with the high osmolarity in marine environments (44). All strains harbored genes encoding phage shock protein (PSP) (Fig. 2b), which may reduce the cellular energy status and increase the competition by stabilizing the cellular membrane and protecting them from extracytoplasmic stress (45). Similar to the two genera *Liberibacter* (46) and *Thalassobius* (47), the two-component regulatory system chv as a basic stimulus-response coupling mechanism was identified in the genus *Qipengyuania* (Fig. 3b), which enables them to sense early and quickly respond to sudden environmental changes (48). The type II and type IV secretion systems (T2SS and T4SS) were overrepresented in the genus *Qipengyuania* (Fig. 2b), which would enable them to more efficiently obtain essential nutrients from ocean environments with limited resources and endows them with enhanced flexibility and adaptability to multiple stresses (49–51).

**The genus *Qipengyuania* is characterized by carotenoid production.** The genomic comparison demonstrated that the clustering patterns of genes involved in carotenoid biosynthesis were similar in the genus *Qipengyuania* (Fig. 4a). The three key genes *crtB*, *crtI*, and *crtY* encoding phytoene synthase, phytoene desaturase, and lycopene β-cyclase in turn formed a cluster in each genome. In this cluster, an unusual gene (*log*) encoding a putative protein that appeared to be a homologue of LOG involved in cytokinin biosynthesis in bacteria (52) was located between the *crtB* and *crtI* genes. Surprisingly, a predicted gene (*hyp*) encoding a hypothetical protein occurred immediately downstream of the *crtI* gene in six strains (Table S5); the phylogenetic and similarity analyses showed that the *hyp* gene sequences were not homologous in six strains

(Fig. 4b), thereby indicating that this gene may not function in carotenogenesis. Unlike the cluster of the *crtB*-log-*crtI*-*crtY*, other essential genes, including *crtE*, *crtG*, *crtZ*, and *crtW* encoding polyprenyl synthetase, 2,2′-*β*-hydroxylase, *β*-carotene hydroxylase, and *β*-carotene ketolase, in turn, were scattered in their genomes. The most conspicuous feature of the *crtG* gene in the genus *Qipengyuania* was its colocation with another gene *DUF2141*, which belongs to the DUF2141 superfamily with unknown function (53). The *crtW* gene was located immediately upstream of the *crtZ* gene in 10 strains.

Four distribution patterns of genes related to carotenoid biosynthesis were determined in this genus, including types I, II, III, and IV (Fig. 4a). Intriguingly, the colony color with types I and II was yellow, while that with types III and IV was orange (Fig. 4c). According to the presumptive pathway of carotenoid biosynthesis (Fig. S6) in the genus *Qipengyuania*, yellow-colored strains most likely accumulated zeaxanthin, caloxanthin, and nostoxanthin, while orange-colored strains may contain zeaxanthin, adonixanthin, and erythroxanthin. Furthermore, the absorption spectra of the extracted pigments exhibited typical spectral characteristics of carotenoids with absorption bands in the region between 400 and 570 nm. The obvious difference was that two absorption peaks at 454 and 482 nm were detected in 21 yellow strains, while a wide absorption spectrum spanning 450 to 480 nm was identified in 10 orange strains (Fig. 4d). Notably, the relative quantities of the different carotenoids and their isomers may result in differences in color type and intensity (17), but the carotenoid separation and identification were beyond the scope of this study. As a result, the genus *Qipengyuania* was characterized by carotenoid production based on genome-wide characterization and pigment determination.

**Some *Qipengyuania* bacteria may be aerobic anoxygenic photoheterotrophs.** Contrary to prevalent carotenoids in the genus *Qipengyuania*, the photosynthesis genes appeared in only the two strains 6B39[T] and 6D47A[T]. The arrangement and composition of the two photosynthesis gene clusters (PGC) were almost identical (Fig. 4e), indicating that they should come from a common ancestor. The PGC organization in the two strains comprised two conserved subclusters, *bchIDO-crtCDF-bchCXYZ-pufBALM* and *bchFNBHLM-lhaA-puhABC-acsF-puhE-hemA*. Strain 6B39[T] had the *pufC* gene encoding the photosynthetic reaction center subunit M, but strain 6D47A[T] did not, suggesting that the *pufC* gene may be inessential for photosynthetic growth (54). The PGC arrangement in the two strains belonged to type III (forward *crtF-bchCXYZ-puf* plus forward *bchFNBHLM-lhaA-puh*) according to the direction and order of genes (55), which was documented in the genus *Erythrobacter* (56). Since the two genetic markers, the *pufM* and *bchY* genes, are frequently used to infer the phylogeny of anoxygenic phototrophs, the PufM- and BchY-based phylogenetic analyses of strains 6B39[T] and 6D47A[T] and reference strains were conducted in this study. The phylogeny demonstrated that most strains sharing the close relationships with strains 6B39[T] and 6D47A[T] were phylogenetically affiliated with the two genera *Novosphingobium* and *Erythrobacter* (Fig. 4f and g). As a result, it was proposed that the PCGs in strains 6B39[T] and 6D47A[T] may be acquired from closely related genera through HGT. It has been proven that bacterial strains to take up foreign genetic material such as DNA and RNA from relatives much more easily than from unrelated strains (57). However, the currently putative HGT event awaits further confirmation in the following study. Based on these analyses, a putative bacteriochlorophyll *a* biosynthesis pathway was proposed in the two strains (Fig. S7). The photosynthetic pigment was checked in batch cultures of the two isolates and control strain under light and dark conditions. Contrary to that of the control strain, the characteristic peak of the bacteriochlorophyll *a* (17) was not detected in *in vivo* absorption spectra for the two strains (Fig. S8), which may result from its low production. Unexpectedly, the light seemed to promote the growth of the two strains and positive strain (Fig. S9). As a result, the two strains should be members of aerobic anoxygenic photoheterotrophs based on the presence of photosynthesis genes and the growth promoted by light.

**The 15 novel *Qipengyuania* species are identified in this study.** The putative novel species determined by genome-wide analyses were identified using a polyphasic taxonomic approach. Most strains were rod or short rod shaped (Fig. S10), while a few strains were pleomorphic. Among 16 isolates, only 4 had polar or lateral flagella (Fig. S10),

matching the presence of flagellar biosynthesis-related genes in their genomes (Fig. 2b). Especially unexpectedly, it was discovered for the first time that a few strains appeared to produce outer membrane vesicles and/or their analogues (Fig. S10), but their biological functions and biogenesis in the genus *Qipengyuania* remain to be resolved. Colonies of all strains were orange or yellow, as shown in Fig. 4c and Fig. S11. The growth ranges of temperature, pH, and NaCl and their optima for all isolates were similar to those of reference type strains (Table S6). Some isolates and type strains were observed to be grown at 4 and 10°C, showing cold adaptability. All strains could grow under heterotrophic conditions, for example, the nutrient-rich MA medium. Almost all strains could hydrolyze Tweens 20, 40, and 60, some strains could hydrolyze Tween 80, and a few strains could hydrolyze starch, cellulose, and skimmed milk (Table S6). In the API ZYM and API 20NE tests, all strains presented similar physiological and biochemical characteristics. However, as shown in Table S6, Fig. S10 and S11, and the species description below, some morphological and physiological characteristics distinguish these strains from other known species, such as positive activities for $\beta$-glucuronidase, $\beta$-glucosidase, and arginine dihydrolase in strain GH1[T] and for urease in strains 1NDH10[T] and GH1[T].

Chemotaxonomy is an important indicator in microbial polyphasic taxonomy (58). Therefore, the chemotaxonomic features of all isolates were conducted and compared in parallel with those of reference type strains. The compositions of cellular fatty acids for all isolates were almost consistent with those of 15 type strains, with summed feature 8 ($C_{19:1}$ $\omega7c$ and/or $C_{18:1}$ $\omega6c$) as the major component (Table S7) and supporting the allocation into the genus *Qipengyuania*. However, the proportions of some fatty acids varied considerably in these strains, such as $C_{16:0}$, $C_{17:1}$ $\omega6c$, and summed feature 3 ($C_{16:1}$ $\omega7c$ and/or $C_{16:1}$ $\omega6c$). The polar lipid profiles of all isolates were similar to those of type strains, including diphosphatidylglycerol, phosphatidylethanolamine, and phosphatidylglycerol as common components (Table S6 and Fig. S12). An obvious difference was that two components of phosphatidylcholine and sphingoglycolipid were detected in most strains, while only one of two components was found in some strains. The respiratory quinones of all isolates were identified as ubiquinone-10, which was in accordance with reference type strains of the genus *Qipengyuania*. These analyses demonstrated that all isolates present consistent chemotaxonomic traits characterized by the genus *Qipengyuania* and should be classified into the genus *Qipengyuania*.

**Conclusions.** In summary, the present study significantly enriched the strain and genetic resources of the genus *Qipengyuania* through isolation from various coastal habitats and genome sequencing. Whole-genome-based phylogeny of the genus *Qipengyuania* implied well-supported phylogenetic relationships and demonstrated a high degree of genetic diversity in combination with dDDH and ANI analyses. Comparative genomic analysis revealed genomic variation and an open pangenome model of the genus *Qipengyuania* that indicates the ability to acquire new genes. Genomic and physiological analyses demonstrated versatile metabolic capacities such as carotenoid production, giving the genus *Qipengyuania* adaptive advantage and great application potential. This study provided sufficient genotypic and phenotypic data, which could differentiate 15 novel species from the known species of the genus *Qipengyuania*, expanding the number of described species to almost double, and we propose the names *Qipengyuania xiapuensis* sp. nov., *Qipengyuania xiamenensis* sp. nov., *Qipengyuania gelatinilytica* sp. nov., *Qipengyuania vesicularis* sp. nov., *Qipengyuania sphaerica* sp. nov., *Qipengyuania aurantiaca* sp. nov., *Qipengyuania polymorpha* sp. nov., *Qipengyuania aestuarii* sp. nov., *Qipengyuania huizhouensis* sp. nov., *Qipengyuania psychrotolerans* sp. nov., *Qipengyuania intermedia* sp. nov., *Qipengyuania proteolytica* sp. nov., *Qipengyuania aerophila* sp. nov., *Qipengyuania qiaonensis* sp. nov., and *Qipengyuania mesophila* sp. nov. (Table 1).

## MATERIALS AND METHODS

**Bacterial isolation and culture.** In this study, the bacterial strains were isolated using a standard dilution and plating method. Considering the nutritional heterogeneity of coastal samples, eutrophic and oligotrophic media were used to isolate bacteria, including marine agar 2216 (MA; BD), 0.1× Marine agar 2216 (0.1 MA), Reasoner's 2A (R2A; Haibo, Qingdao), and improved R2A prepared by filter

**TABLE 1** The proposed 15 novel species of the genus *Qipengyuania* in this study

| Proposed species | Type strains |
| --- | --- |
| *Qipengyuania xiapuensis* | 1NDW9$^T$ = GDMCC 1.2378$^T$ = KCTC 82662$^T$ |
| *Qipengyuania xiamenensis* | 1XM1-15A$^T$ = GDMCC 1.2379$^T$ = KCTC 82610$^T$ |
| *Qipengyuania gelatinilytica* | 1NDH1$^T$ = GDMCC 1.2372$^T$ = KCTC 82606$^T$ |
| *Qipengyuania vesicularis* | 1NDH10$^T$ = GDMCC 1.2373$^T$ = KCTC 82663$^T$ |
| *Qipengyuania sphaerica* | GH29$^T$ = GDMCC 1.2371$^T$ = KCTC 82661$^T$ |
| *Qipengyuania aurantiaca* | 1NDH13$^T$ = GDMCC 1.2375$^T$ = KCTC 82607$^T$ |
| *Qipengyuania polymorpha* | 1NDH17$^T$ = GDMCC 1.2376$^T$ = KCTC 82608$^T$ |
| *Qipengyuania aestuarii* | GH1$^T$ = GDMCC 1.2370$^T$ = KCTC 82605$^T$ |
| *Qipengyuania huizhouensis* | YG19$^T$ = GDMCC 1.2369$^T$ = KCTC 82604$^T$ |
| *Qipengyuania psychrotolerans* | 1XM2-8$^T$ = GDMCC 1.2380$^T$ = KCTC 82611$^T$ |
| *Qipengyuania intermedia* | GH38$^T$ = GDMCC 1.2368$^T$ = KCTC 82603$^T$ |
| *Qipengyuania proteolytica* | 6B39$^T$ = GDMCC 1.2364$^T$ = KCTC 82599$^T$ |
| *Qipengyuania aerophila* | GH25$^T$ = GDMCC 1.2366$^T$ = KCTC 82601$^T$ |
| *Qipengyuania qiaonensis* | 6D47A$^T$ = GDMCC 1.2365$^T$ = KCTC 82600$^T$ |
| *Qipengyuania mesophila* | YG27$^T$ = GDMCC 1.2367$^T$ = KCTC 82602$^T$ |

sterilization (59). More specifically, 1 g mangrove soil and coastal sediments from an aquaculture pond and tidal flats was added into 9 mL sterile deionized distilled water. These mixtures were placed in a shaker overnight at 160 rpm at 28°C and then plated on isolated media. After the incubation at 28°C for a week, colonies were picked out based on colony morphology and then streaked onto the new media. Although numerous isolates were recovered from these samples, only isolates related to the genus *Qipengyuania* were used in this study, that is, other isolates were excluded. Given that all isolates and reference type strains were grown well on MA medium, their phenotypic tests were performed on MA medium or in marine broth (MB, BD) under the same conditions unless otherwise specified.

**16S rRNA gene sequencing and phylogenetic analysis.** Genomic DNA of each isolate was extracted from fresh cells grown on the MA medium using the HiPure Bacterial DNA kit (Magen Biotech Co., Ltd.) following the manufacturer's instructions. The 16S rRNA gene was amplified by PCR using universal bacterial primers 27F and 1492R (60) with the PCR MasterMix (G-clone [Beijing] Biotech Co., Ltd.) in a 50 $\mu$L PCR system. PCR products were sequenced using amplification primers at Suzhou Genewiz Biotechnology Co., Ltd. The sequences were assembled using the software DNAMAN version 8 (Lynnon Biosoft, www.lynnon.com). The 16S rRNA gene sequences of related taxa were obtained from the EzBioCloud (www.ezbiocloud.net) (61) and NCBI databases. Multiple alignments of sequences were performed using the software MAFFT version 7.037 with default parameters (62). Pairwise similarities of 16S rRNA gene sequences were calculated using DNAMAN. The phylogenetic tree based on 16S rRNA gene sequences was reconstructed using the software IQ-TREE version 2.1.2 (63) based on the maximum likelihood (ML) method (64) under the TVM+F+I+*G*4 nucleotide substitution model, which was selected by ModelFinder (65). Support for the ML tree was inferred by ultrafast bootstrapping with 10,000 replicates (66). The phylogenetic trees of other genes, including *hyc*, *bchY*, and *pufM*, described below were reconstructed using the software MEGA version X (67) with the distance option according to Kimura's two-parameter model and clustering with the neighbor-joining (NJ) method (68) with default parameters and 1,000 bootstrap replicates (66). Type strain *Aurantiacibacter gangjinensis* CGMCC 1.15024 was used as an outgroup in phylogenetic analyses.

**Genome sequencing and annotation.** The genome of each isolate was determined using the Illumina NovaSeq PE150 platform of Shanghai Majorbio Bio-Pharm Technology Co., Ltd. About 1 Gbp clean data of each isolate was generated, almost all reaching 200-fold coverage. The high-quality reads were assembled using the software SPAdes version 3.8.1 with default parameters (69). Genomes of reference type strains were obtained from the GenBank database. The genomic quality was assessed using the software CheckM version 1.0.9 (70), which inspected the existence of gene markers specific to the lineage of the order *Sphingomonadales* (UID3310). For consistency, the gene prediction and genomic annotation were reperformed using the software Prokka version 1.13 (71) and the fully automated Rapid Annotation using Subsystem Technology (RAST) pipeline (72) with default parameters. The functional roles of annotated genes of all strains were assigned and grouped in subsystem feature categories. The key genes and metabolic pathways were determined based on the annotation by RAST.

**OGRI calculation and phylogenomic analysis.** Overall genome relatedness indices (OGRI) including digital DNA-DNA hybridization (dDDH) and average nucleotide identity (ANI) were estimated using the genome-to-genome distance calculator (GGDC) version 2.1 online service with the recommended formula 2 (73) and the software FastANI version 1.31 (74), respectively. The dDDH and ANI values were visualized using the "Heatmap" tool of the software TBtools version 1.0981 (75). The whole-genome phylogenetic analysis of the genus *Qipengyuania* was conducted based on two sets of core genes. The phylogenomic trees were reconstructed based on the below concatenated core genes set by the software IQ-TREE version 2.1.2 with the maximum likelihood (ML) method under the LG+F+R7 nucleotide substitution model and based on the up-to-date 92 bacterial core genes set by the software UBCG version 3.0 with its pipeline and default parameters (24), respectively. The bootstrap analysis with 1,000 replicates was performed to determine the reliability of branches obtained from the two phylogenomic trees (66).

**Comparative genomic analyses of the genus *Qipengyuania*.** The bacterial pangenome analyses tool (BPGA) pipeline (76) was used to perform a pangenome analysis with default parameters. The pangenome size of the genus *Qipengyuania* was fitted into a power-law regression function $f(x) = a \times x^b$ with a built-in program

of the BPGA, in which $f(x)$ was the total number of gene families, $x$ stood for the number of tested genomes, and $b$ was a free parameter. If the exponent $b$ was less than zero, the pangenome of the genus *Qipengyuania* was suggested to be "closed." In this case, the size of the pangenome was relatively constant, even if new genomes were added into the analysis. On the contrary, the pangenome was suggested to be "open" in the case that $b$ was between zero and one. The size of the core genome of the genus *Qipengyuania* was fitted into an exponential decay function $f1(x) = c \times e^{(dx)}$ with a built-in program of the BPGA, in which $f1(x)$ stood for the number of core gene families, while $c$ and $d$ were free parameters. Orthologous genes were identified with the USEARCH algorithm using a threshold of 0.5. Variations of the similarity threshold to 0.3, 0.4, 0.6, and 0.7 did not significantly alter the number of gene families; therefore, the default threshold of 0.5 was chosen. Pangenome/core genome plots were calculated over 500 iterations. Core, accessory, and unique genes, and genes encoding carbohydrate active enzymes (CAZymes), were functionally annotated using the eggNOG mapper v2 (77). The data of the pangenomes were visualized using the base R graphics unless specified otherwise.

**Phenotypic, chemotaxonomic, and pigment characterization.** Cellular and colony morphology, motility, Gram staining, and anaerobic growth of isolates were performed according to the methods described previously (78). Catalase and oxidase activities, the ranges and optima of temperature, pH, and NaCl, and the growth on different media were performed according to the methods described previously (79). The autotrophic growth was determined according to the method by Knittel et al. (80) and using the strain *Thiomicrospira* sp. LY2 (unpublished data) as a positive control. Additional biochemical tests were carried out using API ZYM and API 20NE strips (bioMérieux) according to the manufacturer's instructions, with the modification of adjusting the NaCl concentration to 3.0% in all tests. For the fatty acid analysis, cells were harvested from the third quadrants on the MA medium. Cells were saponified, methylated, and extracted using the standard MIDI (Sherlock Microbial Identification System, version 6.0B) protocol. The cellular fatty acids were analyzed by gas chromatography (Agilent Technologies 6850) (81). The respiratory quinones and polar lipids were analyzed according to the method described previously (82). Pigment detection was conducted following the method described previously (83). The effect of light on the growth of strains with photosynthesis genes was determined using the strain *Erythrobacter longus* OCh 101$^T$ (GDMCC 1.491$^T$) as a positive control, which is a model microorganism for the study of aerobic anoxygenic photosynthesis.

**Data availability.** The 16S rRNA gene and genome sequences generated in this study have been deposited in the GenBank database under accession numbers provided in Tables S1 and S2.

## SUPPLEMENTAL MATERIAL

Supplemental material is available online only.
**SUPPLEMENTAL FILE 1**, PDF file, 3.6 MB.

## ACKNOWLEDGMENTS

We thank Aharon Oren (Hebrew University of Jerusalem, Israel) for checking the proposed names and etymologies of the new species and Jianyang Li (Third Institute of Oceanography, Xiamen, People's Republic of China) and Kunpeng Huang (Evergreen feed industry, Fuzhou, People's Republic of China) for the sample collection.

We declare that the research was conducted in the absence of any commercial or financial relationships that could be construed as a potential conflict of interest.

This article does not contain any studies with human participants or animals performed by any of the authors.

This work was jointly supported by the Natural Science Foundation of China (32170118), the Science and Technology Planning Projects of Guangdong Province (2021B1212050022 and 2019B030316017), and the GDAS' Project of Science and Technology Development (2020GDASYL-20200103017).

Conceptualization: Y.L.; data curation: Y.L. and T.P.; formal analysis: Y.L.; funding acquisition: Y.L., M.-R.D., and H.Z.; investigation: Y.L., T.P., and J.D.; methodology: Y.L., T.P., and J.D.; project administration: H.Z. and M.-R.D.; resources: M.-R.D. and H.Z.; software: Y.L.; Supervision: Q.Y. and H.Z.; validation: Y.L. and T.P.; visualization: Y.L.; writing - original draft: Y.L.; writing - review & editing: Y.L. and H.Z.

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
