## [Reviewer comments · Microbiology Spectrum]

Microbiology Spectrum

Comparative Genomics Reveals Genetic Diversity and Metabolic Potentials of the Genus *Qipengyuania* and Suggests Fifteen Novel Species

Yang Liu, Tao Pei, Juan Du, Qing Yao, Ming-Rong Deng, and Honghui Zhu

Corresponding Author(s): Honghui Zhu, Institute of Microbiology, Guangdong Academy of Sciences

Review Timeline:

Submission Date:	August 24, 2021
Editorial Decision:	October 13, 2021
Revision Received:	November 20, 2021
Editorial Decision:	February 6, 2022
Revision Received:	March 15, 2022
Accepted:	April 7, 2022

Editor: Gaurav Sharma

Reviewer(s): Disclosure of reviewer identity is with reference to reviewer comments included in decision letter(s). The following individuals involved in review of your submission have agreed to reveal their identity: Feng Cai (Reviewer #2); Xuewei Xu (Reviewer #3)

Transaction Report:

DOI: <https://doi.org/10.1128/Spectrum.01264-21>

October 13, 2021

Prof. Honghui Zhu

Guangdong Provincial Key Laboratory of Microbial Culture Collection and Application,
State Key Laboratory of Applied Microbiology Southern China,
Guangdong Open Laboratory of Applied Microbiology,
Guangdong Microbial Culture Collection Center (GDMCC),
Institute of Microbiology, Guangdong Academy of Sciences, Guangzhou 510070, China
Guangzhou, Guangdong 510070
China

Re: Spectrum01264-21 (Comparative Genomics Analysis of the Genus *Qipengyuania* Reveals Extensive Genetic Diversity and Metabolic Versatility, Including the Description of Fifteen Novel Species)

Dear Prof. Honghui Zhu:

Thank you for submitting your manuscript to Microbiology Spectrum. This is a well-executed project that will provide several useful insights for the scientific community. Now we have received the comments from both reviewers for your submitted manuscript. Both reviewers have recommended additional experiments and explanations. Therefore, I would like to invite you to revise your manuscript in light of the referees' comments. Your manuscript needs major revisions and the comments from the referees are appended below for your attention. Please ensure that the added explanations and comments in the rebuttal letter are well explained throughout the text and supported by clearly described methods.

When submitting the revised version of your paper, please provide (1) point-by-point responses to the issues raised by the reviewers as file type "Response to Reviewers," not in your cover letter, and (2) a PDF file that indicates the changes from the original submission (by highlighting or underlining the changes) as file type "Marked Up Manuscript - For Review Only". Please use this link to submit your revised manuscript - we strongly recommend that you submit your paper within the next 60 days or reach out to me. Detailed information on submitting your revised paper are below.

Link Not Available

Sincerely,

Gaurav Sharma

Editor, Microbiology Spectrum
<https://sites.google.com/view/sharmaglab/>

Journals Department
Reviewer comments:

Reviewer #1 (Comments for the Author):

The manuscript entitled "Comparative Genomics Analysis of the Genus *Qipengyuania* Reveals Extensive Genetic Diversity and Metabolic Versatility, Including the Description of Fifteen Novel Species" by Liu et al describes and compares newly identified bacteria from the genus *Qipengyuania*.

The manuscript is of interest and combines many approaches to investigate the phylogeny, the genomics, the morphology, and the functional potential of new isolates.

Major comments:

1-The presented results come together very nicely and the narrative is supporting the main findings. However, based on the provided pieces of evidence I found the introduction section misleading. Indeed, the introduction section is highlighting the extreme diversity, the functional potential, and the biotechnological applications of (some) marine microbes but this only vaguely translates into the result section. Based on this study, the Qipengyuania genus needs some (suggested) revision however, there is very little information about how this group might be of biotechnological interest or adapted to "adverse conditions", etc.

2-The manuscript is extremely long. I would recommend streamlining the result section by combining the results that all point in the same direction (e.g., comparative genomics, phylogeny) and focus on the key difference between the identified strains in the main text providing an accompanying table with the discriminating factors only. The rest (i.e., detailed strains description) should be provided as supplementary data.

3-The discussion about pan vs. core vs. accessory genes could be shorter. Indeed, besides the fact that the pan-genome is open, there is little-to-no useful information about the number of genes in this category A vs. category B ...

4-The figures are very nice, in high resolution, on the screen... I don't know how they will look once in the "printed" format. I would consider re-designing the figures for the main text to focus on the key (discriminating) aspects of the study and provide the "complete" high-res figures as supplementary data.

Minor comments: (mostly about the wording/style/English)

I75. This is true for all the microbes - not just the marines ones!

I77. what is a high salt concentration? Are authors talking about hyper-saline environments?

I77. these bacteria are from coastal environments ... I am not sure if it fits into the definition of oligotrophic environments? If it is irrelevant for this paper, don't mention it!

I80. I disagree with this. Can you provide some references for this statement? Some marine microbes are very interesting, but not all, and these are definitely not the only ones!

I82. How much is "recent decades" ? that doesn't sound very recent.

I89. what time?

95. need rephrasing

I96. No, there are many references older than that (e.g., Snell et al. 1999)

I107-111 is confusing? This sentence is missing something!

I114. I have never heard about this family -so if you could provide some references supporting this.

I126. what is subterrestrial sediment?

I127. this is not global - it is almost exclusively coastal and/or sediment!

I130. There is definitely some diversity in enzymes/phenotypes but is there any reference that actually shows some (real) biotechnological application for these bacteria, their enzymes, and metabolites? If not this kind of statement is irrelevant. I don't like the statement that suggests the because it is new, it has to be biotechnologically interesting.

I148. can we get a table with coordinates and environment for each sample?

I158. you don't mean colonial -> colony

I162. What is "good growth". Can you please elaborate?

I175. references and acc.?

I176. Replace under -> with

I264. most unpublished data? Can you please elaborate?

I277. being found doesn't make it prevalent! in addition, the isolation is really based on cultivation ... thus there is no real data about the "prevalence" rather it is about the occurrence!

I302. differences highlight differences ... this title does not look good.

I310. "wider" than what?

I319. can you comment on the difference in genes vs. cds? I guess these are the rRNA, tRNA...

...

I470. the genomic diversity ... their distinct survival strategies in. Or may not. Maybe they just don't have the exact same evolutionary history. Can you please connect some specific traits to specific niches? Do you have data for the fitness of these bacteria across environments?

I484. Acclimation strategy is really about the physiological response of organisms to various environmental conditions. it is not the same as evolution, although they are connected. Thus are you talking about acclimation or acquisition of the pathway? Can you please elaborate?

I491. In general, the lack of traits does really inform on the functional potential of the characterized strains. I am not sure why there is a paragraph about this. Btw CBMs are non-catalytic domains generally attached to other (catalytic domains), not proteins. This suggests that you are counting domains, not proteins, this would need some clarification?

I507. All the microbes, not just the marine ones.

l549. If I am correct, there is no data about the fitness of the bacteria to various conditions, environments (?). The initial environment of each strain is not characterized either! Thus this whole paragraph is highly speculative - I recommend tuning it down and moving it to the supplementary data.

l715. remove powerful

l729 -> all the protologues should be provided in supplementary data.

Reviewer #2 (Comments for the Author):

Major comments:

The authors of Liu et al. presented a comparative genomic study on 15 putatively new Qipengyuania species and the reference ones. This genus harbors 13 known species which are all whole-genome sequenced (WGS) so far. Thus, the WGS data of these 15 new species are of great importance and interest adding to this genus. However, I wonder how the manuscript with its current version helps us to understand the diversity, taxonomy, ecology, genomics, and evolution of this marine Proteobacteria. The manuscript does not read like classical taxonomic research nor a comparative genomic study which may be mainly due to inadequate data mining. The genomic part somehow reads like a company report offering a short summary of the generated genome. Neither of the conclusions at the taxonomic or genomic levels are meaningful. Please find my main concerns as follows.

1. How authors have assigned a new species to this genus? Which species concept was applied, GCPSR? What are the criteria regarding the morphological, phylogenomic/phylogenetic, and polyphasic features that distinguish each species? If pairwise similarity was also used, what was the similarity (%) threshold adopted? Hardly a new species can be assigned if only a supported branch in phylogeny is shown without mentioning the species concept or criteria. Such well-separated (supported) branches/even clades can result from insufficient sampling. More sequences (strains) can always add to the "clustering" on phylogeny namely putative new species. Moreover, if I did not miss something in the ms, how many strains or isolates were examined for each species? One per each would not be sufficient to describe a species. Therefore, I doubt the taxonomic conclusions here.

2. The conclusion (Line 279-300) on the insufficiency and low resolution of 16S rRNA distinguishing bacterial species are known. Therefore it is not necessary to strengthen it here.

3. There is a lack of "biological novelty" in the present study. If the authors have mined out some genetic features (genotype, e.g., CAZyme, pigmentation, etc.) for these bacteria, then the corresponding phenotype characterization (wet lab evidence) should be followed. Therefore, I would strongly suggest the authors to give more space to the pigmentation and industrial application of this bacterium in the Introduction, which would lead the readers to the focus (the main biological novelty outcome) of the present study. Whereas, some descriptive data can be weakened or "hidden" in supplements.

4. The phylogenomic trees are useful. However, I doubt the correctness of the clade assignment by the authors. A consistent criterion should be applied. Thus the "so-called" Group IV should be ideally divided into three groups, otherwise, the outgroup will be included in this group (IV) as well.

5. Table 1 can be integrated into Fig. 2, otherwise, it can be shown in supplements rather than in the main text.

6. The text of the ms requires extensive editing and proofreading. Please find several examples regarding this issue in the minor comments below.

7. The main conclusions are pale, which needs an in-depth digging of the generated data. Ideally, an integration of morphology and genotypes to the ecology of the bacteria would add meaningful "content" to the current study.

Minor comments:

1. Please rewrite the title. It is lengthy and confusing with its present version.

2. line 33, check the grammar correctness of this sentence.

3. Add a few words after "industrial application" to give examples for this point.

4. line 36, "determined their genome sequences"? Please rewrite this sentence.

5. line 38, delete "but".

6. line 41, identify the abbreviations in the first place mentioning "dDDH and ANI".

7. line 45-48, please do not fuse several sentences into one by adding an "and". It causes confusion.

8. line 48, I doubt the "novelty" of these 15 novel species, as you did not give that much biological "novelty" on them.

9. line 51-52, delete the text "and will ... mechanism".

10. line 76, on the earth

11. line 77, cyclings
12. line 126, Table S1, while data are shown in the Introduction part?
13. line 139, functional potential? Do you mean industrial application potential?
14. line 182-183, while to use NJ (MEGA) here?
15. line 332, change "accordant" to "concordant"
16. "among the 16 isolates"

Please note, this is not an exhaustive list, serious proofreading is still needed.

Staff Comments:

Preparing Revision Guidelines

Please return the manuscript within 60 days; if you cannot complete the modification within this time period, please contact me. If you do not wish to modify the manuscript and prefer to submit it to another journal, please notify me of your decision immediately so that the manuscript may be formally withdrawn from consideration by Microbiology Spectrum.

Reviewer comments:

Reviewer #1 (Comments for the Author):

The manuscript entitled "Comparative Genomics Analysis of the Genus *Qipengyuania* Reveals Extensive Genetic Diversity and Metabolic Versatility, Including the Description of Fifteen Novel Species" by Liu et al describes and compares newly identified bacteria from the genus *Qipengyuania*.

The manuscript is of interest and combines many approaches to investigate the phylogeny, the genomics, the morphology, and the functional potential of new isolates.

Reply: We sincerely thank you for the positive evaluation and kind suggestions concerning our research and manuscript. These comments are all valuable and very helpful for revising and improving our manuscript, as well as the important guiding significance to our researches. We have studied comments carefully and have made correction which we hope meet with approval. The point-by-point responses are as follows:

Major comments:

1-The presented results come together very nicely and the narrative is supporting the main findings. However, based on the provided pieces of evidence I found the introduction section misleading. Indeed, the introduction section is highlighting the extreme diversity, the functional potential, and the biotechnological applications of (some) marine microbes but this only vaguely translates into the result section. Based on this study, the *Qipengyuania* genus needs some (suggested) revision however, there is very little information about how this group might be of biotechnological interest or adapted to "adverse conditions", etc.

Reply: Thanks for the positive evaluation and kind comments from you.

We strongly agree with the comment for the introduction. The contents of the introduction in the original manuscript are not closely related to the theme of this study. Therefore, according to your advice, we have reorganized the structure of the introduction and rewritten the related contents by adding detailed information on the taxonomy, ecological distribution, and some functional features of bacteria within the genus *Qipengyuania*. The details of the modification in the introduction are as follows:

The genus *Oipengyuania*, belonging to the family *Erythrobacteraceae* (1) of the order *Sphingomonadales* within the class *Alphaproteobacteria*, was first described by Feng et al. in 2015 with the accompany of the proposal of type species *Oipengyuania sediminis* (2) using a polyphasic taxonomic approach. The genus *Oipengyuania* was initially identified as Gram-staining-negative, facultatively aerobic, and chemoheterotrophic bacteria. In August 2020, a new taxonomic framework of the family *Erythrobacteraceae* was established based on the phylogenomic reconstruction of core genes and genomic similarity analyses (3). Based on the up-to-date taxonomic profile, ten species of the genus *Erythrobacter* was proposed to be transferred into the genus *Oipengyuania* (3). Soon after, the species *Erythrobacter flavus* was also

reclassified as *Oipengyuania flava* (4). Just recently, a new species *Oipengyuania soli* was described along with a proposal for the reclassification of “*Erythrobacter mangrovi*”, “*Erythrobacter aureus*” and “*Erythrobacter nanhaiensis*” into the genus *Oipengyuania* (5). The genus *Oipengyuania* currently comprises 13 species with valid names (<https://lpsn.dsmz.de/genus/qipengyuania>) and 3 species with effectively published names to be validated. After dissecting these species descriptions, it was found that the genus *Oipengyuania* is physiologically and phylogenetically heterogeneous. However, the genome-based genetic diversity and phylogeny of the genus *Oipengyuania* have not been well studied, to some extent due to the lack of enough representative strains and genome sequences available.

Bacteria of the genus *Oipengyuania* have been frequently isolated from various habitats. The type strain of the species *O. sediminis* was obtained from the subterrestrial sediment of Qiangtang Basin in Qinghai-Tibetan plateau, P. R. China (2); other all type strains were isolated from various marine environments, such as seawater (6, 7), estuary water (8), intertidal and deep-sea sediments (9, 10), starfish (11), and mangrove soil (5, 12). These results showed that the genus *Oipengyuania* may be worldwide spread in coastal environments, like many other taxa for example the *Roseobacter* lineage (13). In recent years, the genus *Oipengyuania* has attracted increasing attention due to great potential applications in agriculture, biotechnology, and industry, such as the inhibition of the growth of *Fusarium oxysporum* (14) and seven harmful alga (15), the production of a halotolerant thermoalkaliphilic esterase (16), sulfur-containing carotenoids (17), and poly-beta-hydroxybutyrate (18), the retardation of the corrosion of 2205 duplex stainless steel in marine conditions by the formation of biofilm (19), and the oxidation of thiosulfate to zero-valent sulfur (20). Several genomic analyses of the genus *Oipengyuania* have provided some vital insights into their psychrophilic adaptations (21) and functions in the biogeochemical cycles of phosphorus (22) and sulfur (20). However, we are still far away from a comprehensive understanding of metabolic commonalities and discrepancies between different species of the genus *Oipengyuania*, due to the lack of genomic comparisons.

To gain a better appreciation for the interspecies genomic diversity and metabolic breadth of the genus *Oipengyuania*, 16 new isolates were recovered from coastal samples and sequenced. And then, we conducted a comparative genomic investigation (pan-genome) of the genus *Oipengyuania* to explore its phylogenetic relatedness and industrial application potential. The results suggest that while the primary metabolic pathways are well conserved (e.g., the central carbohydrate metabolism and ammonia assimilation), certainly predicted metabolisms and genome features were markedly different than what is known from this genus. This study provides insight into some unifying features and ancillary pathways of the genus *Oipengyuania* that may contribute to their survival in marine environments.

We sincerely hope that the modified introduction can provide as much information as possible on the research progress and challenges of the genus *Oipengyuania* to the

editor, reviewers, and readers without a literature review.

2-The manuscript is extremely long. I would recommend streamlining the result section by combining the results that all point in the same direction (e.g., comparative genomics, phylogeny) and focus on the key difference between the identified strains in the main text providing an accompanying table with the discriminating factors only. The rest (i.e., detailed strains description) should be provided as supplementary data.

Reply: Thanks for your kind comment.

According to the kind suggestions from you, we have tried our best to refine the contents of the manuscript. The number of words in the text and in the protologue of the manuscript was reduced by about 1500 and 1000, respectively. Table 1 in the original manuscript was put in supplementary materials. More details of manuscript reduction are shown in the responses below.

3-The discussion about pan vs. core vs. accessory genes could be shorter. Indeed, besides the fact that the pan-genome is open, there is little-to-no useful information about the number of genes in this category A vs. category B.

Reply: Thanks for your kind comment.

According to the kind suggestions from you, we have refined the pan-genome analysis and related discussion in the latest version of the manuscript as follows:

A total of 15,172 gene families was identified in the pan-genome analysis. The core genes, accessory genes, and unique genes respectively contained 1,146 (7.6%), 5,283 (34.8%), and 8,743 (57.6%) gene families (Fig. 3a). The percentages of core genes, accessory genes, and unique genes in each genome varied considerably (Table S3), probably indicating a heterogeneous pattern of genomic diversity and evolution. The curves of pan-genome and core-genome sizes indicated an open pan-genome of the genus *Oipengyuania*, which was supported by the parameter b value (0.651913) in the power-law regression function (Fig. 3b). That is, new unique genes would further increase the pan-gene pool size with the increase of the genome number (Fig. S4). In general, an open pan-genome is predominant in bacteria that are susceptible to horizontal gene transfer (HGT) (53). Based on the power-law regression model used in this study, similar results were determined in the species *Enterococcus faecalis* (54) and *Acinetobacter baumannii* (55) and the genera *Metallosphaera* (56) and *Corynebacterium* (57).

4-The figures are very nice, in high resolution, on the screen... I don't know how they will look once in the "printed" format. I would consider re-designing the figures for the main text to focus on the key (discriminating) aspects of the study and provide the "complete" high-res figures as supplementary data.

Reply: Thanks for the positive evaluation on these figures and kind suggestions.

According to the kind suggestions, the original Figures 1 and 2 were put in the supplementary materials as Figures S1 and S2. The re-designed Figure 1 has been shown in the revised manuscript.

Minor comments: (mostly about the wording/style/English)

175. This is true for all the microbes - not just the marines ones!

177. what is a high salt concentration? Are authors talking about hyper-saline environments?

177. these bacteria are from coastal environments ... I am not sure if it fits into the definition of oligotrophic environments? If it is irrelevant for this paper, don't mention it!

180. I disagree with this. Can you provide some references for this statement? Some marine microbes are very interesting, but not all, and these are definitely not the only ones!

182. How much is "recent decades" ? that doesn't sound very recent.

189. what time?

95. need rephrasing

196. No, there are many references older than that (e.g., Snell et al. 1999)

1107-111 is confusing? This sentence is missing something!

Reply: Thanks for nine comments above.

The introduction in the original manuscript has been reorganized and rewritten. The related contents have been deleted in the revised manuscript. Therefore, the above nine comments have no longer responded at here. Thanks for your understanding.

1114. I have never heard about this family -so if you could provide some references supporting this.

Reply: Thanks for these kind comments.

According to your suggestion, we have added a suitable reference (DOI: 10.1007/978-3-642-30197-1_376), which provides an overall view of the family *Erythrobacteraceae*, such as the taxonomy, phenotypic and genomic analyses, isolation and ecological distribution, and application.

To the best of our knowledge, the two most distinctive features of the family *Erythrobacteraceae* are aerobic anoxygenic photosynthesis and carotenoid production. The first aerobic anoxygenic phototrophic bacteria is *Erythrobacter longus* OCh 101, which belongs to the family *Erythrobacteraceae*. Since then, aerobic anoxygenic phototrophs have become a hot topic of scientific research.

With this in mind, the two aspects of the genus *Qipengyuania* were focused on in our study, including the identification, phylogenetic analysis, and arrangement pattern of genes and biosynthetic pathways for carotenoids and bacteriochlorophyll *a*, and the detection of two pigments.

1126. what is subterrestrial sediment?

Reply: Thanks for the comment.

The subterrestrial sediment was collected from a borehole at a depth of 228.0 m, located in the Qiangtang Basin of Qinghai-Tibetan plateau, P. R. China (DOI: 10.1099/ijsem.0.000472). We have detailed the source of the subterrestrial sediment in the latest version of the manuscript.

1127. this is not global - it is almost exclusively coastal and/or sediment!

Reply: Thanks for the comment. Changed accordingly.

1130. There is definitely some diversity in enzymes/phenotypes but is there any reference that actually shows some (real) biotechnological application for these bacteria, their enzymes, and metabolites? If not this kind of statement is irrelevant. I don't like the statement that suggests the because it is new, it has to be biotechnologically interesting.

Reply: Thanks for the comment.

We have added more detailed descriptions and references that show some important functions of the genus *Qipengyuania* with application potential. We have revised related contents in the latest version of the manuscript.

1148. can we get a table with coordinates and environment for each sample?

Reply: Thanks for the comment.

The detailed information on sample sources, types, and coordinates have been shown in Table S2.

1158. you don't mean colonial -> colony

Reply: Thanks for the comment. Changed accordingly.

1162. What is "good growth". Can you please elaborate?

Reply: Thanks for the comment.

We would be happy to give you a detailed explanation. In the present study, the growth of all isolates and reference type strains on five media including R2A, MA, nutrient agar (NA), Luria-Bertani (LB) agar, and tryptic soy agar (TSA) were tested by using the streaking method according to the described method (DOI: 10.1099/ijsem.0.004634). After two weeks of incubation, the growth of all strains was assessed by observing the thickness of the bacterial lawn with naked eyes. Generally speaking, the thicker the bacterial lawn, the better we believe the growth of the strain. By comparison, we determined that the best growth of all isolates and reference strains was observed on the MA medium.

In fact, the current research directly cited the corresponding reference (DOI: 10.1099/ijsem.0.004634), which provide the details about the growth tests. At the same time, in order to describe more clearly, we have revised this sentence in the latest version of the manuscript as follows: Given that all isolates and reference type

strains were grown well on MA medium, their phenotypic tests were performed on MA medium or in Marine Broth (MB, BD) under the same conditions unless otherwise specified.

l175. references and acc.?

Reply: Thanks for the comment.

The related accession numbers have been listed in Table S2.

l176. Replace under -> with

Reply: Thanks for the comment. Changed accordingly. We also revised the same wording in the whole manuscript.

l264. most unpublished data? Can you please elaborate?

Reply: Thanks for the comment.

Before starting with this research, to investigate the diversity and biogeography of the aerobic anoxygenic phototrophic bacteria in coastal regions in P. R. China, more than 1000 isolates were recovered from diverse coastal samples. Among these isolates, only 16 isolates related to the genus *Qipengyuania* were used in this study. Considering that the vast majority of isolates have not been reported, we thus described it as “most unpublished data”.

To describe more clearly, we have rephrased this sentence in the latest version of the manuscript as follows: Before starting with this research, to investigate the diversity and biogeography of the aerobic anoxygenic phototrophic bacteria in coastal regions in P. R. China, more than 1000 isolates were recovered from diverse coastal samples (most unpublished data). Among these isolates, only 16 isolates related to the genus *Qipengyuania* were used in this study.

l277. being found doesn't make it prevalent! in addition, the isolation is really based on cultivation ... thus there is no real data about the "prevalence" rather it is about the occurrence!

Reply: Thanks for the comment.

We fully agree with your comment. We have revised the description with “*Qipengyuania* often occurs in coastal areas” and “the genus *Qipengyuania* seems to occur frequently in coastal areas” in the latest version of the manuscript.

l302. differences highlight differences ... this title does not look good.

Reply: Thanks for the comment.

We have revised the title with “Genomic features of *Qipengyuania* reflect the genetic diversity”.

l310. "wider" than what?

Reply: Thanks for the comment.

We have revised the title with "the genomic sizes of these strains showed a wide range from...".

l319. can you comment on the difference in genes vs. cds? I guess these are the rRNA, tRNA...

Reply: Thanks for the comment.

From our point of view, the predicted genes were comprised of all CDS and RNA genes.

l470. the genomic diversity ... their distinct survival strategies in. Or may not. Maybe they just don't have the exact same evolutionary history. Can you please connect some specific traits to specific niches? Do you have data for the fitness of these bacteria across environments?

Reply: Thanks for the constructive comments and suggestions.

In this study, different survival strategies of the *Qipengyuania* strains were proposed based on the analyses of genomes, key genes, and metabolic pathways. As you can see, this study did not connect any specific traits to specific niches. Honestly, we have not found specific traits to specific niches yet, probably owing to limited strains and tests in this study. But we will keep your suggestion in mind and try our best to unlock some interesting specific traits related to specific niches in future research.

l484. Acclimation strategy is really about the physiological response of organisms to various environmental conditions. it is not the same as evolution, although they are connected. Thus are you talking about acclimation or acquisition of the pathway? Can you please elaborate?

Reply: Thanks for the comments.

As you said, the acclimation strategy should be a physiological response of microorganisms to various environmental conditions. The previous study (DOI: 10.3389/fmars.2018.00435.) has indicated that the glyoxylate shunt was proposed to be an important acclimation strategy for marine heterotrophic bacteria that are subjected to Fe-limitation. Therefore, based on the report and the presence of the pathway of the glyoxylate bypass, it was proposed that the acclimation strategy may also be true in the genus *Qipengyuania*. To describe more clearly, we have moderately modified the sentence in the latest version of the manuscript.

l491. In general, the lack of traits does really inform on the functional potential of the characterized strains. I am not sure why there is a paragraph about this. Btw CBMs are non-catalytic domains generally attached to other (catalytic domains), not proteins. This suggests that you are counting domains, not proteins, this would need some clarification?

Reply: Thanks for the comments.

The detailed responses to your concerns are as follows: (1) The analysis of genes

encoding CAZymes aimed to determine the potential of this genus to utilize carbohydrates. (2) For the annotation of genes encoding CAZymes, the protein FASTA (.faa) file for each genome was used in this study. When a protein sequence was annotated to be related to the carbohydrate-binding module, the protein was defined as the CBM-containing protein. Therefore, the counting of CBMs is the number of proteins rather than the number of domains in this study.

We sincerely hope that our replies can solve your concerns.

1507. All the microbes, not just the marine ones.

Reply: Thanks for the comment. We have revised it.

1549. If I am correct, there is no data about the fitness of the bacteria to various conditions, environments (?). The initial environment of each strain is not characterized either! Thus this whole paragraph is highly speculative - I recommend tuning it down and moving it to the supplementary data.

Reply: Thanks for the comment.

We agree with you that the data about the fitness of the bacteria to various environments is missing and the analysis of this paragraph is highly speculative. According to the kind suggestion from you, we have tried our best to simplify the results and related description of this paragraph.

1715. remove powerful

Reply: Thanks for the comment. Changed accordingly.

1729 -> all the protologues should be provided in supplementary data.

Reply: Thanks for the comment.

This is an important question. To this end, we specially consulted professor Aharon Oren at the Hebrew University of Jerusalem (Jerusalem, Israel), who is a List Editor and former Editor-in-Chief for the International Journal of Systematic and Evolutionary Microbiology at the Society (<https://www.bio.huji.ac.il/en/content/oren-aharon>). A definite answer from him is that if the protologues were provided in supplementary data, the effectively published names of novel species from this study will be not be considered by the list editors of IJSEM for subsequent valid publication, considering that supplementary materials do not fit the criteria of Rule 25a of the International Code of Nomenclature of Prokaryotes concerning a 'permanent record'.

Therefore, all protologues were still shown in the text. Moreover, we have tried our best to simplify the protologues such as the deletion of the data on substrate utilization and enzyme activities from commercial tests of API ZYM and API 20NE.

Reviewer #2 (Comments for the Author):

Major comments:

The authors of Liu et al. presented a comparative genomic study on 15 putatively new Qipengyuania species and the reference ones. This genus harbors 13 known species which are all whole-genome sequenced (WGS) so far. Thus, the WGS data of these 15 new species are of great importance and interest adding to this genus. However, I wonder how the manuscript with its current version helps us to understand the diversity, taxonomy, ecology, genomics, and evolution of this marine Proteobacteria. The manuscript does not read like classical taxonomic research nor a comparative genomic study which may be mainly due to inadequate data mining. The genomic part somehow reads like a company report offering a short summary of the generated genome. Neither of the conclusions at the taxonomic or genomic levels are meaningful. Please find my main concerns as follows.

Reply: We sincerely thank you for the positive evaluation and kind queries on our research and manuscript. These comments are all valuable and very helpful for revising and improving our manuscript, as well as the important guiding significance to our researches. We have studied comments carefully and have made correction which we hope meet with approval. The point-by-point responses are as follows:

1. How authors have assigned a new species to this genus? Which species concept was applied, GCPSR? What are the criteria regarding the morphological, phylogenomic/phylogenetic, and polyphasic features that distinguish each species? If pairwise similarity was also used, what was the similarity (%) threshold adopted? Hardly a new species can be assigned if only a supported branch in phylogeny is shown without mentioning the species concept or criteria. Such well-separated (supported) branches/even clades can result from insufficient sampling. More sequences (strains) can always add to the "clustering" on phylogeny namely putative new species. Moreover, if I did not miss something in the ms, how many strains or isolates were examined for each species? One per each would not be sufficient to describe a species. Therefore, I doubt the taxonomic conclusions here.

Reply: Thanks for the comment.

First of all, we are happy to share our views with you that the new species identification in this study is a classic and standardized research. All tests and methods used in this study have completely followed the guidelines (DOI: 10.1099/ijjs.0.016949-0).

Secondly, for the GCPSR mentioned by the reviewer, we are not sure if it means Genealogical Concordance Phylogenetic Species Recognition (GCPSR). To the best of our knowledge, the GCPSR principle has proven to be a good tool and has been widely used for species delimitation in fungi. But, at present, the identification of bacterial species mainly depends on a polyphasic taxonomic approach including genotype and phenotypic characteristics. That is to say, the identification of fungi and bacteria seems to be significantly different. Therefore, many methods and concepts mentioned by the reviewer in fungi may not be applicable to bacteria.

Thirdly, a 70% digital DNA-RNA hybridization value and 95-96% average nucleotide identity are used as recognized thresholds for bacterial species definition. The two thresholds have been clearly provided in the present manuscript. Meanwhile, the phylogenetic analyses of 16S rRNA gene and genome sequences, morphological, physiological, and biochemical properties, chemotaxonomic features, and genomic characteristics also serve bacterial species definition and description.

Fourth, as you said, the identification and description of almost all new species in this study are based on a single isolate. Honestly, a large number of taxonomic descriptions are still based on a single strain for decades (DOIs: 10.1099/ijms.0.64931-0 and 10.1093/femsle/fnab118). There are several reasons for this situation: (i) the current isolation methods and culture cultivation make it difficult for us to obtain multiple strains belonging to the same species. (2) the accumulation of many strains within a species is a time-consuming process, it may take many years to cultivate multiple strains of a species. For example, after more than ten years of efforts, an Asgard archaeon, Candidatus '*Promethoarchaeum syntrophicum*' was isolated from 25-cm long sediment (DOI: 10.1038/s41586-019-1916-6).

In the current situation, it is obvious that the insistence on species identification based on multiple strains inevitably makes the process difficult and reduces the pace of the identification of novel taxa. As such, a single strain-based description and representation of a novel species is actually an acceptable compromise as it provides at least one cultivated representative to study physiology, ecology, the environmental and clinical significance of the strain.

As a result, this study about the new species description based on a single isolate is acceptable and understandable. The taxonomic conclusions based on solid evidence are reliable.

2. The conclusion (Line 279-300) on the insufficiency and low resolution of 16S rRNA distinguishing bacterial species are known. Therefore it is not necessary to strengthen it here.

Reply: Thanks for the comment.

We agree with you that the 16S rRNA gene with low resolution cannot accurately distinguish closely related strains. Although it seems to be a well-known fact, evidence for the genus *Qipengyuania* is currently insufficient. This study provides solid evidence. Moreover, the 16S rRNA gene cannot accurately differentiate the *Qipengyuania* species, a comprehensive phylogenomics was thus conducted in this study. From the 16S rRNA gene to genomes, the logic of the study is smooth. If the analysis based on the 16S rRNA gene sequences would be deleted, the genome-based analysis will be abrupt. For the two reasons, it would be better that results of the 16S rRNA gene sequences are retained in this study.

At the same time, given the kind comments from you and reviewer 1, the analysis based on the 16S rRNA gene sequences has been simplified in the latest version of the manuscript. The original Figure 1 was put in the supplementary materials as Figure S1.

3. There is a lack of "biological novelty" in the present study. If the authors have mined out some genetic features (genotype, e.g., CAZyme, pigmentation, etc.) for these bacteria, then the corresponding phenotype characterization (wet lab evidence) should be followed. Therefore, I would strongly suggest the authors to give more space to the pigmentation and industrial application of this bacterium in the Introduction, which would lead the readers to the focus (the main biological novelty outcome) of the present study. Whereas, some descriptive data can be weakened or "hidden" in supplements.

Reply: Thanks for the comment.

(1) We fully agree with your kind proposal to give more information about the pigmentation and industrial application of the genus *Qipengyuania* in the Introduction, which was consistent with that from reviewer 1. Following the suggestions from you and reviewer 1, we have reorganized the structure of the introduction and rewritten the related contents by adding detailed information on the taxonomy, ecological distribution, and some functional features of the genus *Qipengyuania*. The details of the modification in the introduction are as follows:

The genus *Qipengyuania*, belonging to the family *Erythrobacteraceae* (1) of the order *Sphingomonadales* within the class *Alphaproteobacteria*, was first described by Feng *et al.* in 2015 with the accompany of the proposal of type species *Qipengyuania sediminis* (2) using a polyphasic taxonomic approach. The genus *Qipengyuania* was initially identified as Gram-staining-negative, facultatively aerobic, and chemoheterotrophic bacteria. In August 2020, a new taxonomic framework of the family *Erythrobacteraceae* was established based on the phylogenomic reconstruction of core genes and genomic similarity analyses (3). Based on the up-to-date taxonomic profile, ten species of the genus *Erythrobacter* was proposed to be transferred into the genus *Qipengyuania* (3). Soon after, the species *Erythrobacter flavus* was also reclassified as *Qipengyuania flava* (4). Just recently, a new species *Qipengyuania soli* was described along with a proposal for the reclassification of "*Erythrobacter mangrovi*", "*Erythrobacter aureus*" and "*Erythrobacter nanhaiensis*" into the genus *Qipengyuania* (5). The genus *Qipengyuania* currently comprises 13 species with valid names (<https://lpsn.dsmz.de/genus/qipengyuania>) and 3 species with effectively published names to be validated. After dissecting these species descriptions, it was found that the genus *Qipengyuania* is physiologically and phylogenetically heterogeneous. However, the genome-based genetic diversity and phylogeny of the genus *Qipengyuania* have not been well studied, to some extent due to the lack of enough representative strains and genome sequences available.

Bacteria of the genus *Qipengyuania* have been frequently isolated from various habitats. The type strain of the species *Q. sediminis* was obtained from the

subterrestrial sediment of Qiangtang Basin in Qinghai-Tibetan plateau, P. R. China (2); other all type strains were isolated from various marine environments, such as seawater (6, 7), estuary water (8), intertidal and deep-sea sediments (9, 10), starfish (11), and mangrove soil (5, 12). These results showed that the genus *Oipengyuania* may be worldwide spread in coastal environments, like many other taxa for example the *Roseobacter* lineage (13). In recent years, the genus *Oipengyuania* has attracted increasing attention due to great potential applications in agriculture, biotechnology, and industry, such as the inhibition of the growth of *Fusarium oxysporum* (14) and seven harmful alga (15), the production of a halotolerant thermoalkaliphilic esterase (16), sulfur-containing carotenoids (17), and poly-beta-hydroxybutyrate (18), the retardation of the corrosion of 2205 duplex stainless steel in marine conditions by the formation of biofilm (19), and the oxidation of thiosulfate to zero-valent sulfur (20). Several genomic analyses of the genus *Oipengyuania* have provided some vital insights into their psychrophilic adaptations (21) and functions in the biogeochemical cycles of phosphorus (22) and sulfur (20). However, we are still far away from a comprehensive understanding of metabolic commonalities and discrepancies between different species of the genus *Oipengyuania*, due to the lack of genomic comparisons.

To gain a better appreciation for the interspecies genomic diversity and metabolic breadth of the genus *Oipengyuania*, 16 new isolates were recovered from coastal samples and sequenced. And then, we conducted a comparative genomic investigation (pan-genome) of the genus *Oipengyuania* to explore its phylogenetic relatedness and industrial application potential. The results suggest that while the primary metabolic pathways are well conserved (e.g., the central carbohydrate metabolism and ammonia assimilation), certainly predicted metabolisms and genome features were markedly different than what is known from this genus. This study provides insight into some unifying features and ancillary pathways of the genus *Oipengyuania* that may contribute to their survival in marine environments.

(2) In this study, we have done our best to correlate genotypic and phenotypic characteristics. For example, polar or lateral flagella were observed in four isolates using transmission electron microscopy, and their flagellar biosynthesis-related genes were determined based on genome-wide comparative analysis. Similarly, the tests of nitrate and nitrite reduction and identification of related genes and metabolic pathways were performed in this study, and the carotenoid pigmentation was also carried out like this. the correlation of more genotypic and phenotypic characteristics was also analyzed in this study, here did not list respectively.

(3) According to your kind suggestion, we have simplified some descriptive results in the latest version of the manuscript and transferred some data to supplementary materials.

4. The phylogenomic trees are useful. However, I doubt the correctness of the clade assignment by the authors. A consistent criterion should be applied. Thus the "so-called" Group IV should be

ideally divided into three groups, otherwise, the outgroup will be included in this group (IV) as well.

Reply: Thanks for the comment.

We agree with what you said that the clades should be assigned based on the consistent criterion. According to your kind suggestion, the Group IV has been re-divided into three groups, which were tentatively designated as Group IV, V, and VI. At the same time, we revised the related descriptions and figure in the latest version of the manuscript.

5. Table 1 can be integrated into Fig. 2, otherwise, it can be shown in supplements rather than in the main text.

Reply: Thanks for the comment.

According to your kind suggestion, we have put Table 1 to the supplementary material as Table S1.

6. The text of the ms requires extensive editing and proofreading. Please find several examples regarding this issue in the minor comments below.

Reply: Thanks for the comment.

According to the kind comments on writing from you and reviewer 1, we have tried our best to revise the English writing of the manuscript carefully. We have also asked two colleagues whose native language is English to revise our manuscript. Therefore, we wish that the modified version of the manuscript will be approved.

7. The main conclusions are pale, which needs an in-depth digging of the generated data. Ideally, an integration of morphology and genotypes to the ecology of the bacteria would add meaningful "content" to the current study.

Reply: Thanks for the comment.

In the current study, 16 new isolates were recovered from various coastal samples and identified as 15 new species using a polyphasic taxonomic approach, expanding the number of described species to almost double. The extensive species diversity and metabolic versatility of the genus *Qipengyuania* were revealed using comparative genomics analysis. More importantly, the morphological observation of cells and colonies, some biochemical tests, the detection of pigment, and other phenotypes can well mirror the analysis of gene and metabolic pathway at the genome level.

Honestly, due to the limited researches of the genus *Qipengyuania* and the lack of environmental parameters and in situ ecological distribution and diversity, the ecology of the bacteria within this genus has never been explored in this study. But we will keep your suggestion in mind and try our best to unlock some interesting specific traits related to specific niches in future research.

We have rephrased the conclusions in the revised version of the manuscript. We wish that the current descriptions in conclusion that can demonstrate the main points and importance of our study will be approved.

Minor comments:

1. Please rewrite the title. It is lengthy and confusing with its present version.

Reply: Thanks for the comment.

According to your kind suggestion, we have revised the title with “Comparative Genomics Reveals Species Diversity and Metabolic Potentials of the Genus *Qipengyuania* and Suggests Fifteen Novel Species”.

2. line 33, check the grammar correctness of this sentence.

Reply: Thanks. We have rephrased this sentence.

3. Add a few words after "industrial application" to give examples for this point.

Reply: Thanks. We have rephrased this sentence.

4. line 36, "determined their genome sequences"? Please rewrite this sentence.

Reply: Thanks. We have rewritten this sentence.

5. line 38, delete "but".

Reply: Thanks. Changed accordingly.

6. line 41, identify the abbreviations in the first place mentioning "dDDH and ANI".

Reply: Thanks. Changed accordingly.

7. line 45-48, please do not fuse several sentences into one by adding an "and". It causes confusion.

Reply: Thanks. We have rephrased this sentence.

8. line 48, I doubt the "novelty" of these 15 novel species, as you did not give that much biological "novelty" on them.

Reply: Thanks for the comment.

We are happy to share our views with you. As we know, novelty can be divided into different levels. From the perspective of microbial taxonomy, the discovery and establishment of new species are to be novel in this study. The biological "novelty" of these new species will be explored in our follow-up study. Thanks for your understanding.

9. line 51-52, delete the text "and will ... mechanism".

Reply: Thanks. Changed accordingly.

10. line 76, on the earth

11. line 77, cyclings

12. line 126, Table S1, while data are shown in the Introduction part?

Reply: Thanks for the comments 10, 11 and 12.

The related contents have been deleted in the revised manuscript. Therefore, the above three comments have no longer responded at here. Thanks for your understanding.

13. line 139, functional potential? Do you mean industrial application potential?

Reply: Thanks. Changed accordingly.

14. line 182-183, while to use NJ (MEGA) here?

Reply: Thanks. For the *bchY* and *pufM* genes, the phylogenetic trees were reconstructed using the software MEGA version X with the neighbor-joining method.

15. line 332, change "accordant" to "concordant"

Reply: Thanks. Changed accordingly.

16. "among the 16 isolates"

Reply: Thanks. Changed accordingly.

Please note, this is not an exhaustive list, serious proofreading is still needed.

Reply: Thanks for the comment.

According to the kind suggestions on the writing from you and reviewer 1, we have tried our best to revise the English writing of the manuscript carefully. We have also asked two colleagues whose native language is English to revise our manuscript. Therefore, we wish that the modified version of the manuscript will be approved.

February 6, 2022

Prof. Honghui Zhu
Institute of Microbiology, Guangdong Academy of Sciences
Xianliezhong Road, 100th
Guangzhou, Guangdong 510070
China

Re: Spectrum01264-21R1 (Comparative Genomics Reveals Species Diversity and Metabolic Potentials of the Genus *Qipengyuania* and Suggests Fifteen Novel Species)

Dear Prof. Honghui Zhu,

Thank you for submitting your manuscript to Microbiology Spectrum. This is a well-executed project that will provide several useful insights for the scientific community. Now we have received the comments from three reviewers for your submitted manuscript. All reviewers have recommended additional experiments and explanations. Therefore, I would like to invite you to revise your manuscript in light of the referees' comments. Your manuscript needs major revisions and the comments from the referees are appended below for your attention. Please ensure that the added explanations and comments in the rebuttal letter are well explained throughout the text and supported by clearly described methods.

Link Not Available

Sincerely,

Gaurav Sharma

Personal Lab Page: <https://sites.google.com/view/sharmaglab/>

Reviewer comments:

Reviewer #2 (Comments for the Author):

The authors have addressed all of the previously raised concerns from my side. I have only a few minor comments/suggestions regarding the text. And ideally, more in-depth mining of the genome data can be provided such that the genomic features can be linked to the ecology and evolution of this bacterium. However, it may depend on the scope of the current ms, and some subsequent research may be followed. as the authors mentioned in the reply letter.

1. There is no "species" in nature, but "speciation". Species is an artificial term that helps us to sort organisms. Therefore, it is not suitable to mention "species diversity" in the ms, but "a high genetic diversity" in this case. Please check it throughout the

ms.

2. line 44, I suppose here it is "phylogenetic relationship" here than "evolutionary relationship".

3. Delete lines 52-53 "the 15 novel...approach" and rephrase the last sentence of the Abstract (lines 53-54).

4. line 89, I suppose here it is "genetically heterogeneous" rather than "phylogenetically heterogeneous". Please be careful with such terms in the text.

5. pls rephrase the sentence of lines 89-92 and put the reason first.

6. line 104, "some harmful alga"? It has to be specified such that harmful to humans, to plants, to what?

Reviewer #3 (Comments for the Author):

The authors have revised this manuscript well according to previous comments. While several points are still needed to be taken into consideration seriously to improve this manuscript.

1. Authors mentioned that the photosynthesis gene cluster was annotated in the strain 6B39T and 6D47AT, while bacteriochlorophyll a was not detected in both strains. They should be more careful to conclude that a few strains were proved to be potentially aerobic anoxygenic photoheterotrophs. It is suggested to carry out determinations of strain growths under light and dark conditions to check whether light promotes their growth or not. Furthermore, a previous study (Xu et al., 2020, Int J Syst Evol Microbiol, 70: 4470-4495) also reported that several Erythrobacteraceae species encode the putative photosynthesis gene cluster did not synthesize bacteriochlorophyll a. It is possible that those gene clusters represent the paralog of aerobic anoxygenic photoheterotrophs, rather than the ortholog of them.

2. Authors mentioned that the phylogenetic trees of other genes were reconstructed using MEGA software (lines 159-161). Which genes did they use?

3. The CO₂ fixation pathway was found in all strains, except for Qipengyuania sediminis CGMCC 1.12928T. While this phenomenon is contradictory with that the genus Qipengyuania members are heterotrophs. Which key genes in the CO₂ fixation were found in them?

Reviewer #4 (Comments for the Author):

This study would be a valuable contribution to the existing literature of the group of the Genus Qipengyuania. In general, the authors have dealt well with all the queries and comments on the last version.

However, I still have one request to the authors to perform further reclassification on the two species "Erythrobacter aureus" and "Erythrobacter mangrovi" as you claimed in lines 309-316. The authors should be done this job by themselves, but not wait for others. Thus, I suggest the authors to add this missing part in the final version.

Other minor comments: Change all the Arabic numbers, such as 15 or 16 to fifteen or sixteen to keep the consistency in the main text and supplementary materials.

Staff Comments:

Preparing Revision Guidelines

Please return the manuscript within 60 days; if you cannot complete the modification within this time period, please contact me. If you do not wish to modify the manuscript and prefer to submit it to another journal, please notify me of your decision immediately so that the manuscript may be formally withdrawn from consideration by Microbiology Spectrum.

Authors have revised this manuscript well according to previous comments. While several points are still needed to be taken into consideration seriously to improve this manuscript.

1. Authors mentioned that the photosynthesis gene cluster were annotated in the strain 6B39^T and 6D47A^T, while bacteriochlorophyll *a* was not detected in both of strains. They should be more careful to conclude that a few strains were proved to be potentially aerobic anoxygenic photoheterotrophs. It is suggested to carry out determinations of strain growths under light and dark conditions to check whether light promotes their growth or not. Furthermore, a previous study (Xu et al., 2020, Int J Syst Evol Microbiol, 70: 4470-4495) also reported that several *Erythrobacteraceae* species encoding the putative photosynthesis gene cluster did not synthesize bacteriochlorophyll *a*. It is possible that those gene clusters represent the paralog of aerobic anoxygenic photoheterotrophs, rather than the ortholog of them.

2. Authors mentioned that the phylogenetic trees of other genes were reconstructed using MEGA software (lines 159-161). Which genes did they use?

3. The CO₂ fixation pathway were found in the all strains, except for *Qipengyuania sediminis* CGMCC 1.12928^T. While this phenomenon is contradictory with that the genus *Qipengyuania* members are heterotrophs. Which key genes in the CO₂ fixation were found in them?

Reviewer comments:

Reviewer #2 (Comments for the Author):

The authors have addressed all of the previously raised concerns from my side. I have only a few minor comments/suggestions regarding the text. And ideally, more in-depth mining of the genome data can be provided such that the genomic features can be linked to the ecology and evolution of this bacterium. However, it may depend on the scope of the current ms, and some subsequent research may be followed. as the authors mentioned in the reply letter.

Reply: We sincerely thank you for the positive evaluation and kind suggestions for the revised manuscript. According to new helpful comments and suggestions from you, we have carefully checked and revised the manuscript which we hope meet with approval. The point-by-point responses are as follows:

1. There is no "species" in nature, but "speciation". Species is an artificial term that helps us to sort organisms. Therefore, it is not suitable to mention "species diversity" in the ms, but "a high genetic diversity" in this case. Please check it throughout the ms.

Reply: Thanks for your kind comment.

Following your kind advice, we have revised the related contents in the latest version of the manuscript.

2. line 44, I suppose here it is "phylogenetic relationship" here than "evolutionary relationship".

Reply: Thanks. Changed accordingly.

3. Delete lines 52-53 "the 15 novel...approach" and rephrase the last sentence of the Abstract (lines 53-54).

Reply: Thanks for the comments.

We have deleted this sentence "The 15 novel...approach". We have also rephrased the last sentence of the Abstract as follows "Collectively, the first insight into the genetic diversity and metabolic potentials of the genus *Oipengyuania* will contribute to better understanding of the speciation and adaptive evolution in natural environments."

4. line 89, I suppose here it is "genetically heterogeneous" rather than "phylogenetically heterogeneous". Please be careful with such terms in the text.

Reply: Thanks. Changed accordingly.

5. pls rephrase the sentence of lines 89-92 and put the reason first.

Reply: Thanks for your suggestion.

We have rephrased this sentence as follows “However, largely due to insufficient representative strains and genome sequences available, the whole-genome-based diversity and phylogeny of the genus *Oipengyuania* have not been well studied to date.”

6. line 104, "some harmful alga"? It has to be specified such that harmful to humans, to plants, to what?

Reply: Thanks. We have detailed the alga as follows “some alga causing red tides”.

Reviewer #3 (Comments for the Author):

The authors have revised this manuscript well according to previous comments. While several points are still needed to be taken into consideration seriously to improve this manuscript.

1. Authors mentioned that the photosynthesis gene cluster was annotated in the strain 6B39T and 6D47AT, while bacteriochlorophyll a was not detected in both strains. They should be more careful to conclude that a few strains were proved to be potentially aerobic anoxygenic photoheterotrophs. It is suggested to carry out determinations of strain growths under light and dark conditions to check whether light promotes their growth or not. Furthermore, a previous study (Xu et al., 2020, Int J Syst Evol Microbiol, 70: 4470-4495) also reported that several Erythrobacteraceae species encode the putative photosynthesis gene cluster did not synthesize bacteriochlorophyll a. It is possible that those gene clusters represent the paralog of aerobic anoxygenic photoheterotrophs, rather than the ortholog of them.

Reply: We sincerely thank you for the positive evaluation and kind suggestions for the revised manuscript. Following your suggestion, we have supplemented related experiments as follows:

First, we tested the effect of light on the growth of strains, including strains 6B39^T, 6D47A^T, and *Erythrobacter longus* OCh 101^T, the latter is regarded as a model microorganism for the study of aerobic anoxygenic photosynthesis. The growth of the three strains in MB medium at 0, 6, 12, 24, 30, 36, and 48 h under light and dark conditions was detected by the OD_{600} absorbance of the culture medium, respectively. As shown in Fig. 1, at the late-logarithmic and stationary phases, the absorbance of strains under light culture was significantly higher than that under dark culture. The results indicated that light can promote the growth of the three strains.

Fig. 1 The growth curves of the three strains under light (L) and dark (D) conditions

Second, we re-detected the photosynthetic pigment of strains 6B39^T and 6D47A^T using strain *E. longus* OCh 101^T as a positive control. The photosynthetic pigment was characterized following the previous method (DOI: 10.1016/j.syapm.2021.126202). As shown in Fig. 1, the characteristic peak of bacteriochlorophyll *a* at $OD_{600} = 760-780$ nm has been detected in strain *E. longus* OCh 101^T under light (L) and dark (D) conditions, but it has not been detected in strains 6B39^T and 6D47A^T.

Fig. 2 The full wavelength absorption spectra of the three strains under light (L) and dark (D) conditions

In conclusion, light can promote the growth of strains 6B39^T and 6D47A^T with complete photosynthesis genes, although the photosynthetic pigments were not detected. Therefore, we prefer to infer that strains 6B39^T and 6D47A^T may be potentially aerobic anoxygenic photoheterotrophs.

At the same time, we have integrated these new results and added related analysis in

the latest version of the manuscript, as shown in the supplementary Figures S8 and S9.

2. Authors mentioned that the phylogenetic trees of other genes were reconstructed using MEGA software (lines 159-161). Which genes did they use?

Reply: Thanks for your kind comment.

We have detailed these genes as follows “other genes including *hyc*, *bchY*, and *pufM* described below”.

3. The CO₂ fixation pathway was found in all strains, except for *Qipengyuania sediminis* CGMCC 1.12928^T. While this phenomenon is contradictory with that the genus *Qipengyuania* members are heterotrophs. Which key genes in the CO₂ fixation were found in them?

Reply: Thanks for your kind comment.

As you said, the phenotype and genotype for the CO₂ fixation may appear to be contradictory. We have checked the related genes related to the CO₂ fixation again. Based on the annotation by the RAST, the four genes respectively encoding glycine cleavage system P2-, P1-, H-, and T-protein are divided into the two subcategories including the “CO₂ fixation” and “Alanine, serine, and glycine” at the same time. In fact, the glycine-cleavage system (GCS) is a multi-component protein system comprising these four proteins above and catalyzing a reversible reaction in terms of glycine cleavage or synthesis. Until now, it functions only in a small group of anaerobic microbes such as *Clostridium acidurici* (DOI: 10.1128/jb.140.2.468-478.1979), *Eubacterium acidaminophilum* (DOI: 10.1111/j.1574-6968.1999.tb13705.x), *Arthrobacter globiform* (DOIs: 10.1016/0003-9861(69)90377-4, 10.1093/oxfordjournals.jbchem.a130483, and 10.1016/0003-9861(76)90236-8), *Candidatus Phosphitivorax anaerolimi* (DOI: 10.1073/pnas.1715549114), and *Desulfovibrio desulfuricans* (DOI: 10.1038/s41467-020-18906-7) which can utilize CO₂, ammonium, 5,10-CH₂-THF and NADH to synthesis glycine. Therefore, it should be should not be responsible for the CO₂ fixation among aerobic *Qipengyuania* bacteria.

At the same time, we tested all *Qipengyuania* strains for autotrophic growth by using autotrophic *Thiomicrospira* sp. as a positive control and the mineral medium containing (g·L⁻¹): K₂HPO₄ 0.5, NH₄NO₃ 0.5, NaCl 0.2, MgSO₄·7H₂O 0.1, CaCl₂·2H₂O 0.05, 1 mL trace element solution (the last three components were added after sterilization) with Na₂S (0.5 mM) and Na₂S₂O₃ (0.5 mM) as electron donors and NaHCO₃ (0.1%, w/v) as a sole carbon source. The results indicated that all strains can not grow by the CO₂ fixation.

As a result, we can conclude from the latest analysis and test results that all *Qipengyuania* strains do not have any CO₂ fixation pathways and are unable to fix

CO₂ under aerobic conditions. That is, all *Qipengyuania* strains are heterotrophic, not autotrophic. We have also revised the related contents and figures in the latest version of the manuscript.

Reviewer #4 (Comments for the Author):

This study would be a valuable contribution to the existing literature of the group of the Genus *Qipengyuania*. In general, the authors have dealt well with all the queries and comments on the last version.

Reply: We sincerely thank you for the positive evaluation and kind suggestions for the revised manuscript. For the comments and suggestions from you, we have carefully checked and revised the manuscript which we hope meet with approval. The detailed responses are as follows:

However, I still have one request to the authors to perform further reclassification on the two species "*Erythrobacter aureus*" and "*Erythrobacter mangrovi*" as you claimed in lines 309-316. The authors should be done this job by themselves, but not wait for others. Thus, I suggest the authors to add this missing part in the final version.

Reply: Thanks for your kind comment.

For the reclassification of the two species "*Erythrobacter aureus*" and "*Erythrobacter mangrovi*", we specially consulted professor Aharon Oren at the Hebrew University of Jerusalem (Jerusalem, Israel), who is a List Editor and former Editor-in-Chief for the International Journal of Systematic and Evolutionary Microbiology at the Society (<https://www.bio.huji.ac.il/en/content/oren-aharon>). His feedbacks are as follows:

The names of the two species are effectively, but not validly, published under the ICNP. Therefore, we can propose the reclassification of the two species in the text, but not in the protologue. This suggestion by Prof. Oren is as described in our current manuscript.

He also suggested that we can email the authors that established the two species and ask them to be validly published the two species in the IJSEM. Honestly, the process is not complicated. The authors only need to send the effectively published PDF file and the two certificates of collection from the two different countries to the office of IJSEM. But unfortunately, our emails were not answered by the authors.

Therefore, we only made a suggestion for the reclassification of the two species into the genus *Qipengyuania* in the current manuscript. Thank you for your understanding for the situation we encountered.

Other minor comments: Change all the Arabic numbers, such as 15 or 16 to fifteen or sixteen to keep the consistency in the main text and supplementary materials.

Reply: Thanks for your suggestion. Changed accordingly.

April 7, 2022

Prof. Honghui Zhu
Institute of Microbiology, Guangdong Academy of Sciences
Xianliezhong Road, 100th
Guangzhou, Guangdong 510070
China

Re: Spectrum01264-21R2 (Comparative Genomics Reveals Genetic Diversity and Metabolic Potentials of the Genus *Qipengyuania* and Suggests Fifteen Novel Species)

Dear Prof. Honghui Zhu,

I am pleased to inform you of the acceptance of your manuscript entitled "Comparative Genomics Reveals Genetic Diversity and Metabolic Potentials of the Genus *Qipengyuania* and Suggests Fifteen Novel Species" in its current form for publication in *Microbiology Spectrum*. I am forwarding it to the ASM Journals Department for publication with this e-mail. You will be notified when your proofs are ready to be viewed.

As an open-access publication, *Spectrum* receives no financial support from paid subscriptions and depends on authors' prompt payment of publication fees as soon as their articles are accepted. You will be contacted separately about payment when the proofs are issued; please follow the instructions in that e-mail. Payment arrangements must be made before your article is published. For a complete list of **Publication Fees**, including supplemental material costs, please visit our website.

Thank you for submitting your paper to *Microbiology Spectrum*.

Sincerely,

Gaurav Sharma
Editor, *Microbiology Spectrum*

Institute of Bioinformatics and Applied Biotechnology (IBAB),
Bengaluru, Karnataka, India
Lab webpage: <https://sites.google.com/view/sharmaglab/>

Reviewer 1: Accept

Reviewer 2: Accept